



# Evaluation of precipitation measurement methods using data from precision lysimeter network

Tobias Schnepper[1, 2, 3], Jannis Groh[1, 4, 5], Horst H. Gerke[4], Barbara Reichert[2], Thomas Pütz[1]

[1]Institute of Bio- and Geoscience IBG-3: Agrosphere, Forschungszentrum Jülich GmbH, Jülich, 52428, Germany
[2]Institute for Geosciences, University of Bonn, Nussallee 8, Bonn, 53113, Germany
[3]GFZ German Research Centre for Geosciences, Telegrafenberg, Potsdam, 14473, Germany
[4]Leibniz Centre for Agricultural Landscape Research (ZALF), Research Area 1 "Landscape Functioning", Working Group "Hydropedology", Eberswalder Straße 84, Müncheberg, 15374, Germany
[5]Institute of Crop Science and Resource Conservation - Soil Science and Soil Ecology, University of Bonn, Nussallee 13,
Bonn, 53113, Germany

*Correspondence to*: Tobias Schnepper (tobias.schnepper@gfz-potsdam.de)

**Abstract.** Accurate precipitation data are essential for assessing the water balance of ecosystems. Methods for point precipitation determination are influenced by wind, precipitation type and intensity and/or technical issues. High-precision weighable lysimeters provide precipitation measurements at ground level that are less affected by wind disturbances and are

assumed to be relatively close to "true" precipitation. The problem in previous studies was that the biases on precipitation data introduced by different precipitation measurement methods were not comprehensively compared and quantified with those obtained by lysimeters under different climatic conditions.

The aim was to quantify measurement errors of standard precipitation gauges as compared to the lysimeter reference and to analyse the effect of precipitation correction algorithms on the gauge data quality. Both correction methods rely on empirical

constants to account for known external influences on the measurements, following a generic and a site-specific approach. Reference precipitation data were obtained from high-precision weighable lysimeters of the TERrestial Environmental Observatories (TERENO)-SOILCan lysimeter network. Gauge types included tipping bucket gauges (TBs), weighable gauges (WGs), acoustical sensors (ASs), and optical laser disdrometers (LDs). The data were collected from 2015–2018 at three sites in Germany and compared with a temporal resolution of 1 hour for precipitation above a threshold of 0.1 mm h$^{-1}$.

The results show that all investigated measurement methods underestimated the precipitation amounts relative to the lysimeter references for long-term precipitation totals with catch ratios (CRs) between 33–92 %. Data from ASs had overall biases of -0.25 to -0.07 mm h$^{-1}$, while data from WGs and LDs showed the lowest measuring biases (-0.14 to -0.06 mm h$^{-1}$ and -0.01 to -0.02 mm h$^{-1}$). Two TBs showed systematic deviations with biases of -0.69 to -0.61 mm h$^{-1}$, while other TBs were in the previously reported range with biases of -0.2 mm h$^{-1}$. The site-specific and generic correction schemes reduced the

hourly measuring bias by 0.13 and 0.08 mm h$^{-1}$ for the TBs and by 0.09 and 0.07 mm h$^{-1}$ for the WGs and increased long-term CRs by 14 and 9 % and by 10 and 11 %, respectively.

It could be shown that the lysimeter reference operated with minor uncertainties in long-term measurements under different climatic conditions. The results indicate that even with well-maintained and professionally operated stations, considerable precipitation measurement errors can occur, which generally lead to a loss of recorded precipitation amounts. Data from

standard precipitation gauges therefore still represent potentially significant uncertainty factors. The results suggest that the application of relatively simple correction schemes, manual or automated data quality checks, instrument calibrations and/or adequate choice of observation periods can help improve the data quality of gauge-based measurements for water balance calculations, ecosystem modelling, water management, assessment of agricultural irrigation needs or radar-based precipitation analyses.





**1 Introduction**

The terrestrial part of the hydrological cycle begins typically with events of precipitation or non-rainfall water. Exact precipitation data are essential to determine the ecosystem water balance within the soil-plant-atmosphere continuum (Porporato and Rodriguez-Iturbe, 2002). To obtain information on local precipitation amounts, point precipitation measurements are carried out using catching and non-catching precipitation gauges (WMO, 2018). At regional and global

scales, precipitation measurement networks connecting a variety of rain gauges provide data to calibrate radar precipitation estimates, climate models and hydrological cycles (Michaelides et al., 2009; Tapiador et al., 2017). Therefore, point precipitation measurements have been intensively studied and several methods for determining precipitation have been developed (WMO, 2018). Depending on the design and functionality of the measuring devices, different errors were identified that affect the precipitation measurements (e.g., Fuchs et al., 2001; Sevruk, 1987; Sevruk and Chvíla, 2005) and therefore also

hydrological models (Bárdossy et al., 2022). Especially the influence of wind has been shown to negatively affect point precipitation measurements (Chvíla et al., 2005; Duchon and Essenberg, 2001; Pollock et al., 2018; Sevruk et al., 1989). The effect depends on wind speed, shielding, the shape of the precipitation gauge and the size, phase and fall velocity of the hydrometeors (Kochendorfer et al., 2017b; Sevruk and Nespor, 1994; Wolff et al., 2015). The use of wind shields (Alter, 1937; Nipher, 1878) can reduce the measuring bias, depending on the shields' design (Kochendorfer et al., 2017a; Watson et al.,

2008; Yang et al., 1999a). In order to quantify erroneous point precipitation measurements and to minimise the influence of wind, ground-level gauges were developed (Goodison et al., 1998; Lanza and Vuerich, 2009; Sevruk and Hamon, 1984). The data obtained from these gauges were used as a reference for comprehensive comparisons of precipitation gauges for solid (Goodison et al., 1998) and liquid precipitation (Lanza and Vuerich, 2009). The comparative studies led to the development of correction methods that are applied during the post-processing of the data (Førland et al., 1996; Richter, 1995; Sevruk,

60  1982).

Weighable lysimeters were recognised in the 1960s as reference system for precipitation gauge comparisons with measurements at the soil surface (McGuiness, 1966; Morgan and Lourence, 1969). Lysimeters were originally developed to determine and quantify soil and plant related processes such as drainage, evapotranspiration, ion exchange, root development and solute transport (Goss and Ehlers, 2009; Hertel and Unold, 2013). A direct determination of precipitation with weighable

lysimeters has been limited in the past by relative coarse temporal resolution (> 10 minutes to daily) and precision. Within the TERrestial Environmental Observatories (TERENO)-SOILCan lysimeter network, high-precision weighable lysimeters are used that are operating with a high temporal resolution (< 1 minute; Pütz et al., 2016). These lysimeters have been successfully used for monitoring precipitation, dew and hoar frost for soil-ecosystems (Gebler et al., 2015; Groh et al., 2018b; Schrader et al., 2013) with a greater resolution and precision than most common gauges (Xiao et al., 2009b).

To assure data quality and reliable accuracy of lysimeter-derived precipitation measurements, Marek et al. (2014) stated, that the pre- and post-processing of lysimeter data were highly important. Time series of lysimeter mass changes are oscillating and temporally autocorrelated (Herbrich and Gerke, 2016). Noise filtering algorithms have therefore been developed since the 2010s to reduce the influence of ambient noise on the data (Hannes et al., 2015; Nolz et al., 2013; Peters et al., 2014; Ross et al., 2020; Vaughan and Ayars, 2009) . The "adaptive window and threshold filter" (AWAT; Peters et al., 2014) executes two

steps to process the data towards a correction of time-variable noise levels (Peters et al., 2017; Peters et al., 2016). This filter solved the problem of underestimating the lysimeter mass change signals at the turning point from precipitation to evapotranspiration or vice versa; it yields an almost unbiased representation of the real signal which is especially important for the quantification of smaller flux rates such as those caused for instance by dew formation (Groh et al., 2018b; Xiao et al., 2009a).

The increased lysimeter data quality (i.e. precision and temporal resolution) has already led to a number of studies that compared lysimeter measurements with those of other precipitation gauges (Gebler et al., 2015; Haselow et al., 2019; Hoffmann et al., 2016; Kohfahl and Saaltink, 2020; Schrader et al., 2013). Gebler et al. (2015) found a 16 % underestimation





of tipping bucket gauge (TB) measured precipitation totals compared to lysimeter data for one year, with 17 % of the difference due to rime and dew and 5.5 % due to fog and drizzle. Snow formation also contributed to the deviations, although the authors

identified problems with snow bridges and snow drift interfering with the lysimeter measurements (Gebler et al., 2015). Haselow et al. (2019) compared precipitation measured with a piezoelectric precipitation sensor, a standard TB and a weighable lysimeter. They found an underestimation by the TB and an overestimation by the piezoelectric sensor compared to the lysimeter measurements. Kohfahl and Saaltink (2020) compared precipitation measurements of a precision lysimeter with two TBs and one weighable rain gauge. The authors found good agreement between the measured data from one TB rain gauge,

the weighable rain gauge and the lysimeter, while recommending correction of the data from the other TB which underestimates precipitation intensities compared to the lysimeter reference.

Within the framework of the TERENO-SOILCan lysimeter network, numerous point precipitation measurement gauges are under operation, producing data with 10-min resolution (Pütz et al., 2016). Up to four different rain gauges are installed in the immediate vicinity of the lysimeter stations and their data can be publicly accessed through the TERENO data platform

(https://ddp.tereno.net/ddp/; for details see "Data availability"; Kunkel et al., 2013). The lysimeter data available within the network is comprehensively processed and tested and has been used for a variety of studies in the field of soil and plant sciences (Groh et al., 2019; Groh et al., 2020a).

For this study, we hypothesise that values derived from hourly precipitation data of high-precision weighable lysimeters can function as reference for a comparison of multiple precipitation measuring methods. To test this, data from multiple lysimeters

installed at respective test sites are graphically compared via scatter plots, a correlation coefficient, and deviations to the arithmetic mean. Furthermore, we hypothesise that if the precipitation measurements of the considered methods are affected by external influences, then the gauge measurements should be on average lower than the ones determined by the lysimeter references for all devices of the same gauge model at all sites. To test this, we compare hourly precipitation amounts as well as long-term precipitation totals determined by these measuring methods with reference data derived from high-precision

weighable lysimeter data via scatter plots and appropriate statistics for all three sites, also considering measuring uncertainties. The final hypothesis is that the use of precipitation correction methods that consider the influence of wind and other typical sources of error on the instruments will reduce the bias between reference and measurements and increase the data quality close to zero. Therefore, two correction methods are applied on the tipping bucket and weighable gauge data during post-processing, and the influence on the bias and overall deviations from the references is studied by comparing the resulting data

with the reference lysimeter-based precipitation.

To conduct this study, high-quality long-term data of four years (2015–2018) from 15 lysimeters, four different types of precipitation measuring methods (11 devices) of three research sites and local weather data with a resolution of 1 hour are available.

## 2 Material and methods

### 115 2.1 Research sites

The data were collected at research sites near Dedelow (DD), Rollesbroich (RO) and Selhausen (SE) in Germany, which are all part of the TERENO-SOILCan network (Fig. 1; Pütz et al., 2016). The TERENO-SOILCan lysimeter network itself belongs to the TERENO project (Zacharias et al., 2011) and comprises twelve different lysimeter research sites in Germany (Pütz et al., 2016). The network purpose is to enable insights into effects of climate change on arable and grassland ecosystems,

including water balance components. To achieve this, a modified space-for-time concept was used, where lysimeters filled with four different soils have been transferred within the TERENO observatories to simulate a time-induced change of climate through a space-induced change of climate for studying effects on crop productivity and soil water and plant nutrient balances (Groh et al., 2020a; Groh et al., 2020b; Pütz et al., 2016). The lysimeters have been used to estimate precipitation and actual





evapotranspiration (Gebler et al., 2015; Groh et al., 2019; Schrader et al., 2013) as well as non-rainfall water that arises from

the formation of dew, hoar frost and rime (Groh et al., 2018b; Meissner et al., 2007).

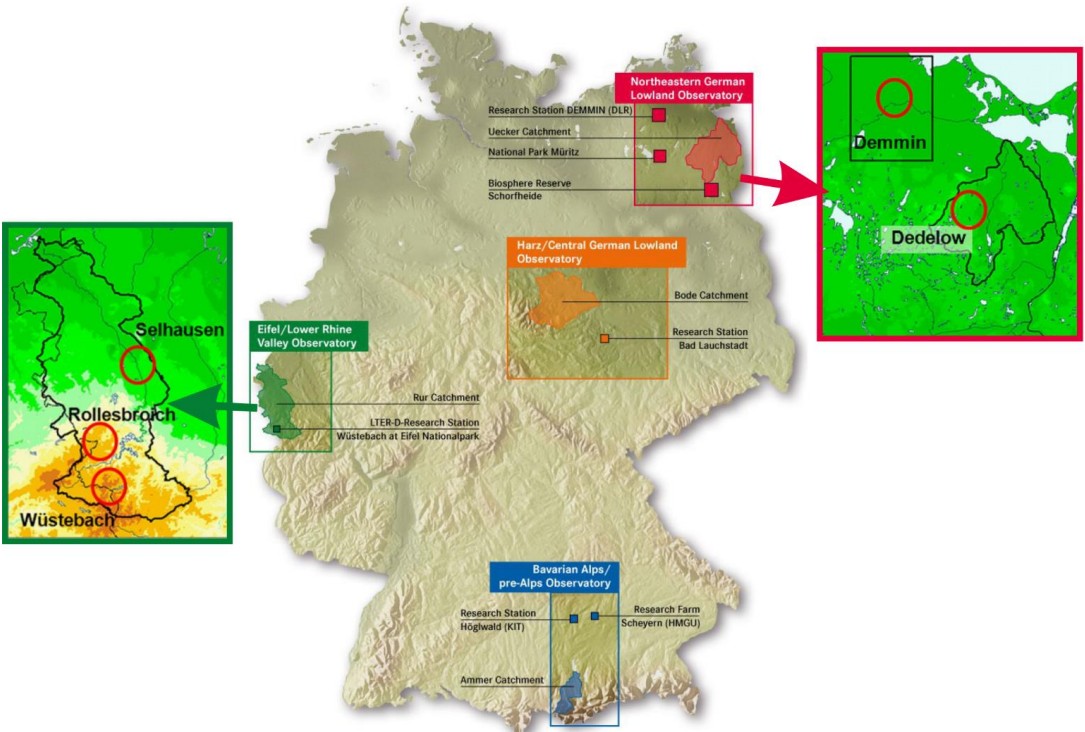

**Figure 1:** The TERrestial Environmental Observatories (TERENO)-SOILCan network (adapted from Pütz et al., 2016). The data for this study was collected at sites near Dedelow (Northeastern German Lowland Observatory), Rollesbroich and Selhausen (both Eifel/Lower Rhine Valley Observatory).

RO, as part of the Eifel/Lower Rhine Valley Observatory, is situated in the northern part of the Eifel, which is a low mountain range in Western Germany (Bogena et al., 2018). The area is widely open and comprises extensively used grassland. The elevation of the site (511 m above sea level) leads to a mean annual precipitation of 1063 mm a$^{-1}$ (DWD, 2021a; 50°38'; 06°12') and a mean annual temperature of 7.7 °C (DWD, 2021b; 50°38', 06°12').

The site in SE is situated in the plain lower Rhine valley and is surrounded by extensively used arable land (Bogena et al.,

2018). It is located in the temperate maritime climate zone with a mean annual temperature of 9.8°C (DWD, 2021b; 50°56', 06°34') and a mean annual precipitation of 723 mm a$^{-1}$ (DWD, 2021a; 51°56', 06°34').

The test site near DD is in the Northeastern German Lowland Observatory of TERENO (Heinrich et al., 2018), approximately 100 km north of the city of Berlin. At the site, a hummocky ground moraine landscape is characterized by lakes and arable land (Heinrich et al., 2018). The lysimeter station is surrounded by arable land. A mean annual temperature of 7.9 °C (DWD,

2021b; 53°19', 13°56') and a mean annual precipitation of 504 mm a$^{-1}$ (DWD, 2021a; 53°17', 13°56') have been determined for the area around the test site in Dedelow.

## 2.2 Equipment and data collection

### 2.1.1 TERENO-SOILCan Lysimeter

The weighable high-precision lysimeters used within the TERENO-SOILCan network are comparable to the "Scientific Field

Lysimeter" described by Unold and Fank (2008). The lysimeters are equipped with soil moisture probes, soil water sampling





devices, tensiometer, a precision weighable system, a thermal flux sensor and silicon carbide porous suction cups (Pütz et al., 2016). Every lysimeter has a surface of 1 m² and a depth of 1.5 m. They are connected to a central service well which contains the power supply, data loggers, measuring transducers, pumps, seepage water tanks including weighable systems, sampling bottles and a data modem (Pütz et al., 2016). For detailed information about the installed sensors and the lysimeter filling, see

Pütz et al. (2016) and Hertel and Unold (2013).

All lysimeters used in this study are of the same type and disposed in lysimeter stations which comprise at least one hexagon of six lysimeters placed around the central service well (Fig. 2 B; Pütz et al., 2016). The weighable systems consist of three load cells (Model 3510, Tedea-Huntleigh, Canoga Parl, CA, USA) with a resolution of 10 g ≅ 0.01 mm precipitation and are mounted at each lysimeters bottom. A gap between lysimeter and the concrete housing is closed with a synthetic resin collar,

which is covered with surrounding soil. The small remaining gap between the collar and the lysimeter is closed with silicone foil fixed at the resin collar, but without any direct contact to the lysimeter vessel to prevent any disturbance on the lysimeter weighing. This ensures a smooth suspension and weighing of the lysimeter (Pütz et al., 2016). The lysimeters were intensively maintained at least every week and the incoming data were daily checked for irregularities and anomalies. To ensure a stable environment for the observations, the type of lysimeter filling remains the same during the observation period. The surrounding

areas were similarly managed as the lysimeters to prevent an "isle effect", that would have affected the determination of precipitation due to exposed vegetation (Hagenau et al., 2015). On the arable lysimeters the crop rotation was winter wheat, winter barley, winter rye and oat during the observation period (2015–2018). No catch crop between the growing seasons was grown. In general, the agricultural activities were in accordance with common agricultural practice (Pütz et al., 2016). The grass lysimeters were cut three to four times per year.

Based on the mass changes of a lysimeter and a seepage water tank, the water fluxes across the soil surface were calculated using the lysimeter water balance (Schrader et al., 2013). The water balance equation for a lysimeter follows:

$$\Delta W = P + NRW - ET_a - D, \tag{1}$$

where $\Delta W$ is the change in the lysimeter mass, $P$ the precipitation, $NRW$ non-rainfall water, $ET_a$ the actual evapotranspiration and $D$ the drainage. The amount of precipitation is calculated from any increase in lysimeter mass after correction with water

fluxes across the lysimeter bottom. It was assumed that no evaporation or transpiration occurred during the specific time period (Schneider et al., 2021; Schrader et al., 2013; Eq. 2). Any decrease of the lysimeter mass change after the correction with fluxes across the lysimeter bottom can thus be related to evapotranspiration if no precipitation occurred during this time interval (Eq. 3).

$$P_i = \begin{cases} WL_i - WL_{i-1} + WT_i - WT_{i-1} > 0 \\ 0 \; for \; WL_i - WL_{i-1} + WT_i - WT_{i-1} \leq 0 \end{cases}, \tag{2}$$

$$ET_i = \begin{cases} -(WL_i - WL_{i-1} + WT_i - WT_{i-1}) \\ for \; WL_i - WL_{i-1} + WT_i - WT_{i-1} < 0 \; , \\ 0 \; for \; WL_i - WL_{i-1} + WT_i - WT_{i-1} \geq 0 \end{cases} \tag{3}$$

where $P_i$ is the precipitation amount during time interval i (kg), $WL$ is the lysimeter mass in terms of the weight (kg), $WT$ is the seepage tank weight (kg) and $ET_i$ is the evapotranspiration amount during time interval i (kg).





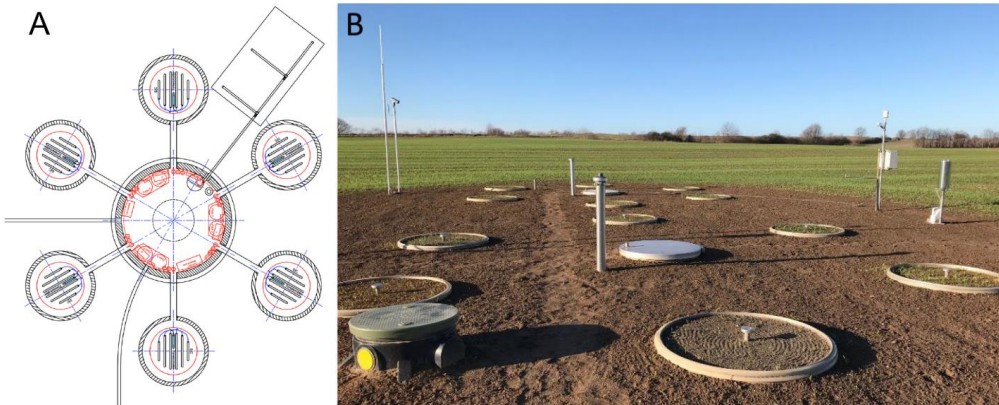

**Figure 2:** A) Technical drawing of the lysimeter hexagon from birds view (from Pütz et al., 2016, provided by UMS AG, Munich, Germany).
B) Two lysimeter hexagons at ZALF´s research station in Dedelow (photo by Hannah Kimmich).

### 2.2.2 Precipitation gauges

Two variants of the same tipping bucket gauge model (TB; "ecoTech Umwelt-Meßsysteme GmbH", Bonn, Germany) are used at the test sites, which differ in the collecting surface of 200 cm² for SE and RO and 400 cm² for DD. The measuring mechanism bases on a funnel with an inlet water pipe which leads the caught precipitation into a tipping bucket, which, when filled with

a certain amount of water, tips to either side of the sharp edge pivot. The motion is determined by the weight imbalance caused by the water dripping from above. The device registers the number of tips with a magnet mounted on the bucket, inducing a signal to a reed switch. A second TB in RO and the TB in DD are additionally equipped with a heating module. The resolution of the TBs is 0.1 mm which indicates an equal volume of one of the buckets compartments.

The weighable gauge (WG) used in this study is the model Pluvio² manufactured by Ott Hydromet, Kempten, Germany. It

uses load cells to determine the differences in weight in the weighing chamber caused by collected precipitation. The gauge has an orifice with a collecting area of 400 cm², a resolution of 0.01 mm and a minimal cumulative precipitation threshold at 60 min collection time of 0.05 mm h$^{-1}$. The device outputs real-time and processed data with a time lag of a few minutes. A heating module is installed to prevent solid precipitation from blocking the gauges inlet. It merely heats the collecting ring around the orifice to avoid artificial loss due to evaporation. The model was used as reference gauge in multiple studies

(Johannsen et al., 2020; Kochendorfer et al., 2017a; Ryu et al., 2016) and was one of the most widely used WGs according to a survey within the WMO members in 2008 (Wong and Nitu, 2010). On the 2017/07/19, a single Alter wind shield was installed around the gauge in SE. This type of shield consists of loose hanging lamellas, which are mounted on a concentric ring around the gauge (Alter, 1937) and is widely established to prevent wind loss and recommended by several institutes (Dengel et al., 2018).

The weather station WXT510 (Vaisala Oyi, Helsinki, Finland) is installed at all sites to measure meteorological standard parameter such as wind speed, wind direction, air temperature, relative humidity, and barometric pressure at 2 m height. The station includes the Vaisala RAINCAP® sensor, an acoustic sensor (AS) with a resolution of 0.01 mm h$^{-1}$ (Salmi and Ikonen, 2005). The AS relies on a piezoelectric detector with a surface of 60 cm², with which the impact of individual raindrops on the surface is measured. Raindrops collide with the surface at their terminal velocity, which is a function of the raindrop diameter.

Depending on the terminal velocity and the mass of the raindrop, every collision creates a particular signal, which is proportional to the volume of the drops, and is then converted to an individual voltage by the piezoelectric detector (Winder and Paulson, 2012). The voltage pulses are filtered, amplified, digitized, and analysed. Knowing the voltage signals per unit time and the surface area, rain duration and -intensity can be determined. Each type of precipitation creates a distinguishable signal which enables the detection of liquid and solid precipitation. Since the terminal velocity of snow is low, the AS can only

differentiate between rain and hail.





The laser disdrometer (LD; Adolf Thies GmbH & Co KG, Göttingen, Germany) is installed on the sites in RO and SE. The instruments use an infrared light-beam with a wavelength of 780 nm and a cross section of approx. 46 cm². It records the reduction of the transmitted light intensity by particles falling through the light-beam and can therefore derive the particle diameter. The strength and duration of the attenuation initiated by the falling particle allows for an inference of its diameter

and velocity, such that the precipitation type, rain and snow, can be distinguished (Bloemink and Lanzinger, 2005; Fehlmann et al., 2020). The LD distinguishes precipitation with a diameter of 0.125 mm and derives the vertical velocity by the measured signal duration (Lanzinger et al., 2006). It offers a low sensitivity threshold with a resolution of 0.001 mm h⁻¹.

Weather data as well as precipitation data from TB, WG and AS are stored on a data logger at DD, RO, and SE (all "envilog Maxi", ecoTech, Bonn, Germany) at 10-minutes intervals (Pütz et al., 2016). Weighing data from the lysimeters are stored in

a data logger (1-minute intervals, DT85, Datataker-Thermo Fisher Scientific Australia Pty Ltd., Scoresby, VIC, Australia), whereas the LD´s data is stored on a local memory card.

**2.3 Data processing**

Lysimeter data, which are available as time series of mass changes in minute resolution, were pre-processed in three steps: i) check of the minutely monitored raw data time series' for plausibility (manually and automatically), ii) application of the

AWAT-filter to reduce the impact of noise on the raw data of the lysimeter mass changes (Peters et al., 2017), and iii) summing the minutely to hourly values; note that hours with missing values were marked as not available (NA).

Weather and gauge precipitation data were similarly treated with a plausibility check (manually and automatically) and an aggregation to hourly values. For this, the precipitation measurements present for 10-minutes intervals were summed, while the 10-minutes measurements from the climate stations such as temperature (°C), wind speed (m s⁻¹) and wind direction (°)

were averaged over one hour.

To determine whether the gauge data then add up to comparable hourly values and to be able to eliminate potential time lags between all devices, representative precipitation events were analysed by comparing the 10-minutes with the hourly resolution (Appendix A1). Since the lysimeter measurements function as reference for the comparison, they are intended to mark the beginning of an event as well as significant peaks within a precipitation event. The comparison revealed a time lag of ten

minutes between lysimeter and gauge data, which is due to time stamps from different data loggers. Another time lag for the WG data was induced by gauge-internal processing. Adjusting the overall gauge data with a 10-minutes inverse time lag and using the raw WG data solved the problem.

**2.4 Reference precipitation values and their uncertainties**

The derivation of a reference precipitation intensity ($P_{ref}$; mm h⁻¹) for the comparison of precipitation data is critical for further

investigations. It was chosen to compute the mean measured precipitation among the lysimeters for each hourly data point (Eq. 4), when at least half of the lysimeters data were available for that hour. This way it is ensured that the mean is not depending on single measurements and that too many data points are discarded because of missing values. At sites (SE and DD) where only three lysimeters of the same soil-ecosystem are operated, the mean is calculated when at least two lysimeters provide measurements:

$$P_{ref} = \begin{cases} \frac{1}{n}\sum_{i=1}^{n} x_i \ for \ n \geq n_{ia}, \\ NA \ for \ n < n_{ia}, \end{cases} \qquad (4)$$

where $n$ is the number of lysimeters providing observations during time interval $i$ (-), $x_i$ the precipitation intensity measured by each lysimeter in the given interval $i$ (mm h⁻¹) and $n_{ia}$ is the number of lysimeters with missing data during time interval $i$ (-).

Following WMO (2018), a rain intensity gauge is required to not exceed defined uncertainties while measuring rain

precipitation intensities ($P_{gauge}$). Under field conditions, an uncertainty of 5 mm h⁻¹ for $P_{gauge}$ < 100 mm h⁻¹ should not be





exceeded. An approach conducted by Vuerich et al. (2009) derives $P_{ref}$ from the mean of four different reference pit gauges. They calculated the uncertainty for the reference values from standard deviations of precipitation measurements from the pit gauges. The authors use the 95 % confidence level (k=2) deducted from two times the standard deviation of the measurements. To adapt the concept, it was decided to proceed as follows: It is assumed that measurements of the same precipitation event

conducted by all available lysimeters are normally distributed. Thus, the standard deviation (SD) between measurements of all lysimeters with the same vegetation cover at each site can be calculated for each hour, where data from all lysimeters is available (Eq. 5). After calculating the lysimeter uncertainties ($U_{ref}$; Eq. 6) each hour is assigned to a category according to its precipitation intensity value, whereby each category comprises a span of 0.1 mm, resulting in 200 categories for $P_{ref}$ of 0 to 20 mm h$^{-1}$, which was the maximum observed precipitation intensity during the observation period across all sites. All uncertainty

values for hours within the respective category are then averaged and the computed mean is attributed back to the respective hours. This assigns an individual uncertainty value to each precipitation intensity interval. The WMO (2018) recommends to add 5 % of $P_{ref}$ to the gauges uncertainties (here lysimeters; $U_{fin}$; Eq. 7) as an upper limit for uncertainties of rain gauges. The final uncertainties are then plotted and smoothed using the "*geom_smooth*" function (R, Package: ggplot2; Wickham, 2016) with cubic splines using the R software (R Core Team, 2020) to provide an uncertainty range for the gauge comparison via

scatter plots.

$$SD = \sqrt{\frac{\sum_{i=1}^{n}(x_i - \bar{x})^2}{n-1}},$$ (5)

$$U_{ref} = P_{ref} \pm 2 * SD,$$ (6)

$$U_{fin} = U_{ref} \pm 0.05 * P_{ref},$$ (7)

with $n$ being the number of observations (-), $x_i$ the precipitation intensities measured by each lysimeter during time interval $i$

(mm h$^{-1}$), $\bar{x}$ the empiric mean of the lysimeter measurements at the site during time interval $i$ (mm h$^{-1}$), $P_{ref}$ the reference precipitation intensity during time interval $i$ (mm h$^{-1}$), $U_{ref}$ the uncertainty of the lysimeter reference during time interval $i$ (mm h$^{-1}$) and $U_{fin}$ the final uncertainty for the gauge comparison for time interval $i$ (mm h$^{-1}$). To additionally reduce potential uncertainties and achieve more comparable results when comparing the measurement methods with the lysimeter reference, only data is used with $P_{ref}$ and $P_{gauge}$ both being $\geq$ 0.1 mm h$^{-1}$, which is the smallest common resolution of all measuring

instruments.

### 2.5 Determination of key parameters

#### 2.5.1 Non-rainfall events

Non-rainfall events (NRE) comprise the formation of dew, hoar frost, and fog. Hourly precipitation data was categorised as NRE, if no rain gauge except the lysimeters or the disdrometer registered precipitation. The disdrometer was excluded because

it could detect fog. Often, combined observations on NREs by leaf wetness (for dew and hoar frost) or visibility sensors (fog) and weighable lysimeters are used to identify NREs (Zhang et al., 2019). No such sensors were available for our sites. Therefore, we followed the approach by Groh et al. (2018b), who restricted the occurrence of NRE to the period between sunset and sunrise without any rainfall detected by WG and the hourly NRE intensity to a maximum rate of 0.07 mm h$^{-1}$ according to assumptions on dew formation by Monteith and Unsworth (2013).

### 2.5.2 Precipitation type and intensity classification

The piezoelectric sensor as well as the optical disdrometer could identify the mentioned precipitation types in a sufficient way (Bloemink and Lanzinger, 2005; Fehlmann et al., 2020; Salmi and Ikonen, 2005). Due to an incomplete data situation, an approach was carried out to derive the precipitation type from the air temperature at the sites. For gauge intercomparisons, temperature ranges of +4 to 0 °C (Gebler et al., 2015), +2.5 to -2.5 °C (Kochendorfer et al., 2017b), +2 to 0 °C (Førland et al.,





1996) and +2 to -2 °C (Ryu et al., 2016) were chosen for mixed precipitation. Here the later threshold of +2°C and -2°C was
used in this study for mixed precipitation (Table 1). Thus, temperatures above 2°C are used to classify liquid precipitation
(rain) and temperatures below -2°C classify solid precipitation (snow). The small number of precipitation events classified as
"snow" can be explained by the pre-filtering of the lysimeter measurements influenced by snow bridges or ice formation. Due
to the small number of "snow" events and the uncertainties involved in quantifying snow and mixed precipitation with
lysimeters and precipitation gauges, hours within both categories are excluded from further investigation.

**Table 1:** Number of hours of three main precipitation types for the years 2015–2018 according to the lysimeter reference. Hours with no
precipitation registered by the lysimeters and non-rainfall events are excluded.

| Site | Rain (h) | Mixed (h) | Snow (h) |
|---|---|---|---|
| **Rollesbroich** | 4695 | 763 | 21 |
| **Selhausen** | 3949 | 186 | 8 |
| **Dedelow** | 2949 | 260 | 4 |

The classification of precipitation intensities was done in accordance to WMO (2018). Here, observations were classified
according to the precipitation registered by the lysimeter reference. Hours with $P_{ref} < 2.5$ mm h$^{-1}$ were regarded as "slight", $2.5
\leq P_{ref} < 10$ mm h$^{-1}$ as "moderate" and $10 \leq P_{ref} < 50$ mm h$^{-1}$ indicated "heavy" rainfall (Table 2).

**Table 2:** Number of hours classified according to the reference precipitation intensity ($P_{ref}$) for the years 2015–2018. Hours with no
precipitation registered by the lysimeters and hours with non-rainfall events are excluded.

| Site | Slight precipitation ($P_{ref} < 2.5$ mm h$^{-1}$) (h) | Moderate precipitation ($2.5 \leq P_{ref} < 10$ mm h$^{-1}$) (h) | Heavy precipitation ($10 \leq P_{ref} < 50$ mm h$^{-1}$) (h) |
|---|---|---|---|
| **Rollesbroich** | 5179 | 269 | 10 |
| **Selhausen** | 3976 | 155 | 4 |
| **Dedelow** | 3073 | 132 | 4 |

**2.6 Correlation analyses**

The precipitation measurements between the reference values and those obtained from the gauges were compared based on x-
y scatter plots (e.g., Haselow et al., 2019; Liu et al., 2013; Yang, 2014). Within a plot, the uncertainty range, the bias (Eq. 8)
as well as the standard deviation of the measurement differences (SDD, Eq. 9), the R²-value (Eq. 10), the number of
observations (n) and the linear regression line (Eq. 11) are given. To enhance the visualisation, additional plots and statistics
for slight precipitation events are provided.

$$Bias = \sum_{i=1}^{n}(X_i - Y_i)/n \,, \tag{8}$$

$$SDD = \sqrt{\frac{\sum_{i=1}^{n}(x_i-y)^2}{n-1}} \,, \tag{9}$$

$$R^2 = \frac{\sum(Y_i-\bar{x})^2 {}^2}{\sum X_i - \bar{x}} \; with \; \bar{x} = \frac{1}{n}\sum_{i=1}^{n}X_i \,, \tag{10}$$

where $n$ is the number of observations (-), $X_i$ the reference precipitation intensity during time interval $i$ (mm h$^{-1}$) and $Y_i$ the
precipitation intensity determined by the precipitation gauge during time interval $i$ (mm h$^{-1}$).

The bias is the average difference between the precipitation measured by the gauge and the reference measurement, so positive
or negative values indicate that the gauges generally over- or underestimated precipitation, respectively. The SDD quantifies
the distribution of the differences and thus the spread of the data points around the 1:1 reference line. The nondimensional
coefficient of determination, R², describes how well the linear regression model, displayed in the plot, fits to the data. Due to
its derivation, it is also closely related to the extent of the spread of the data but does not consider the spread around the 1:1





reference line. The regression lines follow the standard linear regression (Eq. 11) performed in R (R Core Team, 2020), using the "*ggplot2*" package (Wickham, 2016).

$$Y_i = \alpha + \beta X_i,$$ (11)

where $\alpha$ and $\beta$ are regression coefficients.

While interpreting the bias, SDD, R²-value and regression lines, it must be considered that these statistics and methods base on the assumption of normally distributed data. Since precipitation data does not follow such a distribution, low precipitation intensities occur more often than higher intensities and therefore the residuals are heteroscedastic, the statistics might be not enough to describe the data distribution. Therefore, an additional, visual illustration of the data was required, as presented by the scatter plots.

**2.7 Wind error analysis**

To analyse the influence of wind speed at gauge height ($v_g$) on the gauge's precipitation measurements, the catch ratio (CR; Eq. 12) is used. The CR can be described as a function of wind speed and temperature (Goodison et al., 1998; Yang et al., 1998). CRs of different gauges were calculated with regression analyses on the basis of reference measurements at multiple sites (Chen et al., 2015; Goodison et al., 1998; Sugiura et al., 2006; Yang et al., 2000). The CR of a gauge and reference values 335 can also be described with following equation:

$$CR = \frac{P_{gauge}}{P_{ref}} * 100 .$$ (12)

Since the influence of the temperature can be neglected due to the exclusion of mixed and solid precipitation, the primary parameter to consider is the wind speed at gauge height, which can be derived from the measured wind speed at 2 m above ground (Eq. 13; Goodison et al., 1998):

$$v_g = [log(\tfrac{h_g}{z_0}) \, / \, (log(\tfrac{h}{z_0})] * v_h,$$ (13)

where $v_g$ is the wind speed at the height of the gauge orifice (m s⁻¹), $h_g$ the height of gauge orifice above ground (m), $z_0$ the roughness length (0.01 m in winter, 0.03 m in summer), $h$ the height of the wind speed measuring instrument above ground (m) and $v_h$ the wind speed measured by the measuring instrument (m s⁻¹).

**2.8 Precipitation data corrections**

Sevruk (1982) provides a general equation for the correction of precipitation measurements which is adjusted with regard to an improved readability to the notation by Førland et al. (1996; Eq. 14):

$$P_{cor} = k(P_m + P_w + P_e + P_t) ,$$ (14)

with $P_{cor}$ being the corrected precipitation (mm), $k$ the correction factor for wind (-), $P_m$ the measured precipitation (mm), $P_w$ the wetting loss (mm), $P_e$ the evaporation loss (mm) and $P_t$ the loss by trace precipitation (mm).

Few of the correction methods based on Eq. (13) are developed to correct precipitation measurements on an hourly basis, due to the absence or small numbers of automatic gauge types in the last decades. Førland et al. (1996) published the dynamic correction model (DCM) with which Michelson (2004) corrected hourly precipitation data of four different gauge models mainly used in Scandinavian countries. The correction factor for wind loss, k, can thus be adjusted for hourly rain measurements following the DCM (Førland et al., 1996; Michelson, 2004; Eq. 15):

$$k = exp \begin{bmatrix} -0.00101 * ln(P_{gauge}) - 0.012177 * vg * ln(P_{gauge}) + \\ 0.034331 * v_g + 0.007697 + c \end{bmatrix},$$ (15)

where $c$ is the gauge constant (mm 12 h⁻¹).


Michelson (2004) offers empiric constants for wetting loss, evaporation loss as well as the wind induced error for four different gauge models and precipitation types (Table 3-4). The empiric constants apply for 12 h measuring intervals, which can be derived to an hourly basis. Since evaporation does not account for WGs if it is supressed with an oil film (WMO, 2008), only

the gauge constant for determining the wind error has to be chosen for the WG. Michelson (2004) states that precipitation gauge designs, that are similar to the WG used here, have a slightly improved gauge constant compared to the Hellmann gauge (0.0 mm 12 h$^{-1}$), thus a gauge constant of -0.05 mm 12 h$^{-1}$ is chosen for the WG.

To compensate the loss by trace precipitation for the TB, an empiric value provided by Yang et al. (2001) with a loss of 0.1 mm d$^{-1}$ is assumed for days with at least one precipitation event.

**Table 3:** Wetting and gauge constants for Eq. (15) provided by Michelson (2004) for multiple gauge types. SMHI: Gauge used by the Swedish Meteorological and Hydrological Institute.

| Precipitation phase | Constant | SMHI (mm 12 h$^{-1}$) | Hellmann (mm 12 h$^{-1}$) |
|---|---|---|---|
| Liquid | Gauge | -0.05 | 0.00 |
| Liquid | Wetting | 0.07 | 0.14 |

**Table 4:** Daily evaporation loss constants for Eq. (15) as provided by Michelson (2004) for multiple gauge types. SMHI: Gauge used by the Swedish Meteorological and Hydrological Institute.

| Month | Hellmann (mm 12 h$^{-1}$) |
|---|---|
| January | 0.01 |
| February | 0.02 |
| March | 0.03 |
| April | 0.04 |
| May | 0.09 |
| June | 0.15 |
| July | 0.16 |
| August | 0.08 |
| September | 0.02 |
| October | 0.01 |
| November | 0.01 |
| December | 0.01 |


Richter (1995) developed a correction method especially for German Hellmann-type gauges installed at sites, where no wind speed can be determined. He used factors depending on the shielding of the site and the precipitation type, based on long term precipitation measurements. The derived daily correction amount includes all relevant losses (including wind, evaporation, trace, and wetting). To test the potential of the correction method by Richter (1995) to be applied on hourly precipitation data,

a similar approach to the one published by Gebler et al. (2017) is chosen. The authors initially calculated corrections for daily precipitation data according to Richter (1995; Eq. 16 and 17) and then redistributed it to hourly measurements via the ratio of daily measured to daily corrected data. Here it was decided to redistribute the daily corrections via the number of precipitation hours within a day (Eq. 18):

$$P_d^{cor} = P_d + \Delta P_d, \tag{16}$$

$$\Delta P_d = b * P_d^{\varepsilon}, \tag{17}$$

$$P_h^{cor} = \frac{P_d^{cor}}{n_{Pevent}} + P_h, \tag{18}$$

where $P_d^{cor}$ is the corrected daily precipitation (mm d$^{-1}$), $P_d$ the daily precipitation measured by gauge (mm d$^{-1}$), $\Delta P_d$ the amount of daily additional precipitation (mm d$^{-1}$), $b$ the coefficient for the influence of wind exposition at the measurements





site (-), $P_d^\varepsilon$ the empiric coefficient for the precipitation type (-), $P_h^{cor}$ the corrected hourly precipitation (mm h$^{-1}$), $n_{Pevent}$ the

number of hours with $P_{ref} > 0.1$ mm h$^{-1}$ within a day (-) and $P_h$ the hourly precipitation measured by the gauge (mm h$^{-1}$).

## 3 Results

### 3.1 Uncertainty range of the lysimeters

With increasing $P_{ref}$ values, the measurement uncertainties from the lysimeters decrease at all sites. The uncertainty ranges indicate that between 1 and 2.5 mm h$^{-1}$ the relative measuring differences decrease to values below 5 % (Fig. 3). For $P_{ref} > 8$

mm h$^{-1}$, the lack of data does not allow to delineate a clear area of uncertainty. In DD, three rain events with precipitation intensities > 11 mm h$^{-1}$ were associated with increased deviations within the lysimeter measurements, which bias the regression line and therefore the predicted uncertainty range at these precipitation intensities (Fig. 3D).

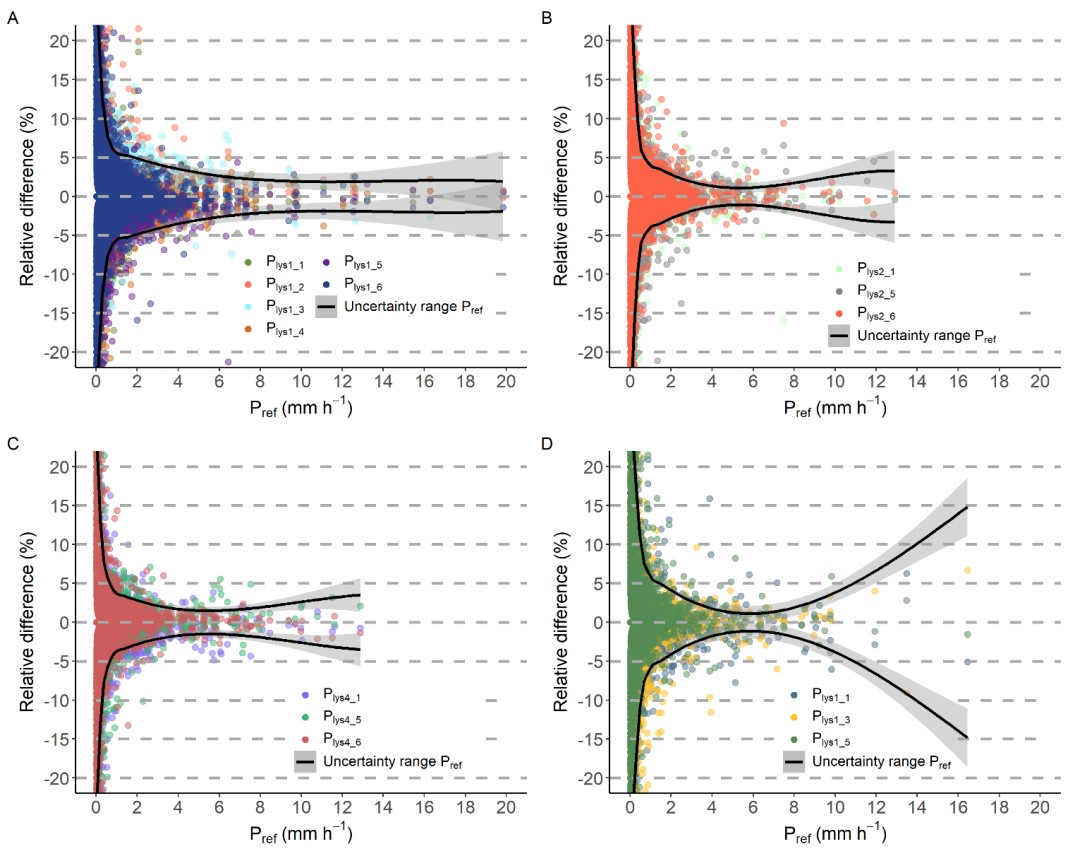

**Figure 3:** Uncertainty ranges of lysimeter references ($P_{ref}$), delimited through solid lines, and differences of each lysimeter measurement
relative to $P_{ref}$. A: Rollesbroich lysimeter station, lysimeter with grassland. B: Selhausen lysimeter station, lysimeter with grassland. C: Selhausen lysimeter station, lysimeter with arable land. D: Dedelow lysimeter station, lysimeter with arable land.

### 3.2 Annual precipitation values

During hours in which all devices at a site were available and at least one gauge (TB, WG or AS) and the lysimeters measured a precipitation intensity of 0.1 mm h$^{-1}$ or higher (Table 5), the TB1 in Rollesbroich (RO) registered 22.9–47.1 % of $P_{ref}$ within
each individual year of operation. The TB in Selhausen (SE) caught 20.4–40.7 %, TB2 (RO) 79.2–83.8 % and the TB in Dedelow (DD) 72.8–84.0 %. The WGs caught 82.2–89.0 % (RO) and 89.4–94.5 % (SE), respectively. The AS measured 62.8–78.2 % (RO), 70.0–82.1 % (SE) and 82.6–91.3 % (DD). The LD (RO) measured 85.7–95.3 %, while the LD (SE) measured



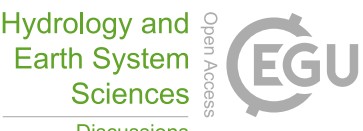

73.6–96.1 %. Deviations between the precipitation totals of the lysimeters with different vegetation cover in SE are considerably low with CRs of 99.8–101.4 %.

**Table 5:** Precipitation sums (P) and catching ratios (CR) of all investigated gauges compared to the references per year. Only hours are considered, where all devices at a site were active, at least one device, tipping bucket (TB), weighable gauge (WG) or acoustic sensor (AS) and the lysimeters did measure a precipitation intensity of at least 0.1 mm h$^{-1}$ and the precipitation is classified as "Rain".

| Site | Year | n | $P_{ref}$ | $P_{lys\_crop}$ | $CR_{lys\_crop}$ | $P_{TB1}$ | $CR_{TB1}$ | $P_{TB2}$ | $CR_{TB2}$ | $P_{WG}$ | $CR_{WG}$ | $P_{AS}$ | $CR_{AS}$ | $P_{LD}$ | $CR_{LD}$ |
|---|---|---|---|---|---|---|---|---|---|---|---|---|---|---|---|
| | | (-) | (mm) | (mm) | (%) | (mm) | (%) | (mm) | (%) | (mm) | (%) | (mm) | (%) | (mm) | (%) |
| Rolles-broich | 2015 | 793 | 709 | | | 334 | 47.1 | 594 | 83.8 | 606 | 85.5 | 499 | 70.4 | 676 | 95.3 |
| | 2016 | 989 | 801 | | | 184 | 22.9 | 634 | 79.2 | 659 | 82.2 | 526 | 65.6 | 687 | 85.7 |
| | 2017 | 1007 | 881 | | | 273 | 30.9 | 732 | 83.1 | 744 | 84.5 | 553 | 62.8 | 795 | 90.2 |
| | 2018 | 694 | 583 | | | 189 | 32.4 | 471 | 80.8 | 519 | 89.0 | 456 | 78.2 | 522 | 89.6 |
| Sel-hausen | 2015 | 736 | 599 | 600 | 100.1 | 244 | 40.7 | | | 549 | 91.7 | 423 | 70.6 | 589 | 98.4 |
| | 2016 | 697 | 554 | 556 | 100.3 | 221 | 39.9 | | | 495 | 89.4 | 388 | 70.0 | 455 | 82.1 |
| | 2017 | 581 | 431 | 430 | 99.8 | 169 | 39.2 | | | 398 | 92.3 | 303 | 70.2 | 317 | 73.6 |
| | 2018 | 548 | 388 | 394 | 101.4 | 79 | 20.4 | | | 367 | 94.5 | 319 | 82.1 | 373 | 96.1 |
| Dedelow | 2015 | 571 | 414 | | | 344 | 83.1 | | | | | 373 | 90.1 | | |
| | 2016 | 641 | 423 | | | 322 | 76.1 | | | | | 351 | 83.1 | | |
| | 2017 | 719 | 693 | | | 504 | 72.8 | | | | | 572 | 82.6 | | |
| | 2018 | 407 | 279 | | | 235 | 84.0 | | | | | 255 | 91.3 | | |

### 3.3 Precipitation intensities

On an hourly basis, all TBs tend to underestimate precipitation relative to the reference (Fig. 4). Measurements of the TBs showed little scattering, indicating consistent measurements with R²-values above or near 0.8 for all devices, except for TB1 in RO (Fig. 4 A2) with R² = 0.53. However, TB1 (RO; Fig. 4 A1) as well as the TB (SE; Fig. 4 C1) showed biases of -0.69 and -0.61 mm h$^{-1}$ and SDDs of 0.90 and 0.85 mm h$^{-1}$ for all rainfall events with $P_{ref}$ ≥ 0.1 mm h$^{-1}$. In hours of intensities between 0.1 to 2.5 mm h$^{-1}$ the biases were -0.48 and -0.44 mm h$^{-1}$ and the SDDs 0.40 and 0.38 mm h$^{-1}$ (Fig. 4 A2 and C2).

Measurements from TB2 (RO; Fig. 4 B1) showed a bias of -0.18 mm h$^{-1}$ and an SDD of 0.31 mm h$^{-1}$ while the TB (DD; Fig. 4 D1) measures with a bias of -0.20 mm h$^{-1}$ and an SDD of 0.41 mm h$^{-1}$. For slight precipitation intensities, measurements from TB2 (RO) had a bias of -0.16 mm h$^{-1}$ and an SDD of 0.20 mm h$^{-1}$ and TB (DD) a bias of -0.14 mm h$^{-1}$ and an SDD of 0.2 mm h$^{-1}$, respectively (Fig. 4 B2 and D2).





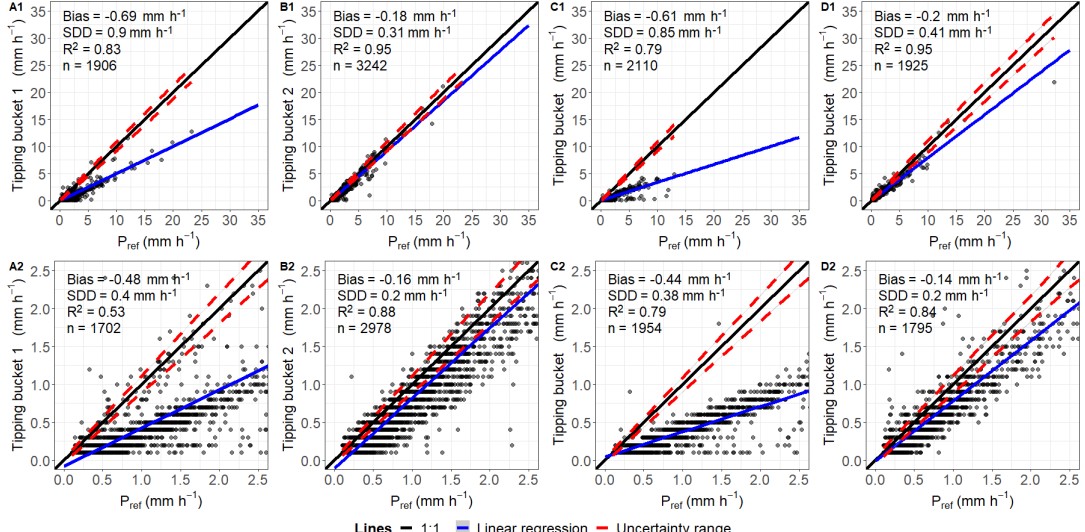

**Figure 4:** Comparison of hourly data determined by tipping buckets and lysimeters ($P_{ref}$) and classified as "Rain". Plots A1–D1 include all measurements taken where both $P_{gauge}$ and $P_{ref}$ are ≥ 0.1 mm h⁻¹. Plots A2–D2 show slight precipitation events and key values are calculated for $P_{ref}$ between 0.1–2.5 mm h⁻¹. A1, A2: Rollesbroich lysimeter station. B1, B2: Rollesbroich EC station. C1, C2: Selhausen lysimeter station. D1, D2: Dedelow lysimeter station.

Measurements conducted by the WG (RO; Fig. 5 A1) showed a bias of -0.15 mm h⁻¹ and an SDD of 0.24 mm h⁻¹. For the time without wind shield, measurements from the WG (SE; Fig. 5 B1) had a bias of -0.08 mm h⁻¹ and an SDD of 0.18 mm h⁻¹, while with shield the bias was -0.04 mm h⁻¹ and the SDD 0.14 mm h⁻¹ (Fig. 5 C1). At $P_{ref}$ ≤ 2.5 mm h⁻¹, the bias decreased from 0.07 to 0.04 mm h⁻¹ and the SDD from 0.16 to 0.12 mm h⁻¹ after the installation of the wind shield (Fig. 5 B2 and C2) in comparison with the WG (RO; Fig. 5 A2) which's measurements for $P_{ref}$ ≤ 2.5 mm h⁻¹ had a bias of -0.11 mm h⁻¹ and an SDD of 0.16 mm h⁻¹.

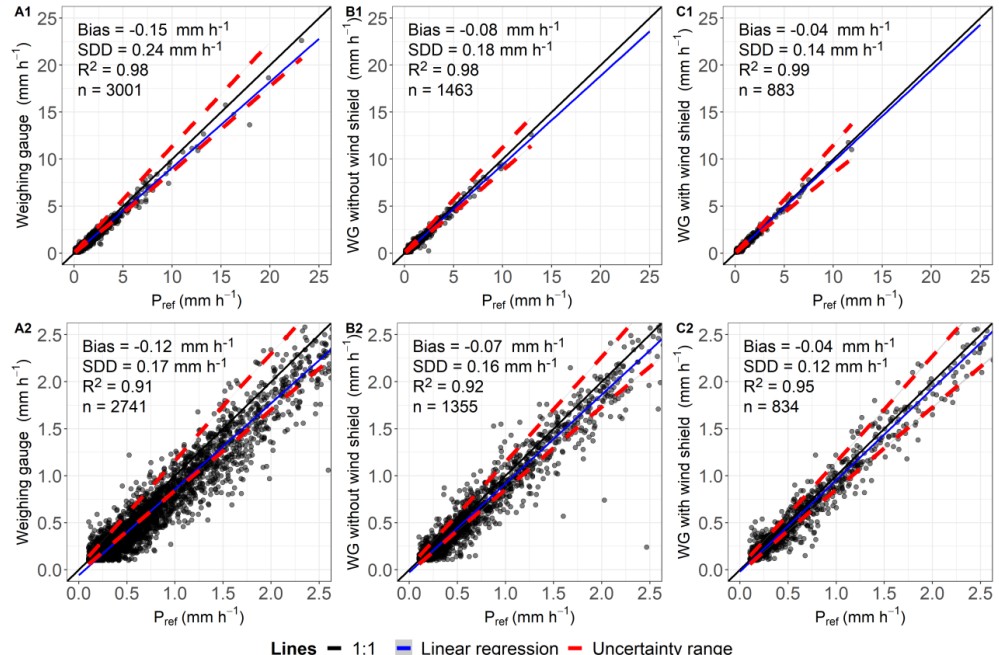

**Figure 5:** Comparison of hourly data determined by weighable gauges and lysimeters ($P_{ref}$) and classified as "Rain". Plots A1–C1 include all measurements taken with both $P_{gauge}$ and $P_{ref}$ are ≥ 0.1 mm h⁻¹. Plots A2–C2 show slight precipitation events and key values are calculated





for $P_{ref}$ between 0.1–2.5 mm h⁻¹. A1, A2: Rollesbroich lysimeter station. B1, B2: Selhausen lysimeter station without installed wind shield around the weighable gauge. C1, C2: Selhausen lysimeter station with installed wind shield around the weighable gauge.

The ASs data scattered largely around the 1:1-line with the tendency of the ASs to underestimate the precipitation intensity (Fig. 6 A1 to C2). Especially for slight precipitation intensities, the measurements correlated poorly with the reference data. To exclude possible errors during the data processing, hourly values were calculated additionally, implementing a positive and negative 10-minutes lag on the data. This led to an even lower degree of agreement with the reference measurements, resulting in higher bias and SDDs than shown in Fig. 6. Additionally, a discrepancy between the measurement intervals from the ASs

and reference would also result in false positives because only sensor data would show precipitation. A plot with the data including $P_{ref} = 0$ (not shown here) reveals no suspicious occurrence of these false positives. Therefore, the data from the ASs was evaluated as valid. Measurements from the AS (RO; Fig. 6 A1) had a bias of -0.25 mm h⁻¹ and an SDD of 0.7 mm h⁻¹. A bias of -0.22 mm h⁻¹ and an SDD of 0.46 mm h⁻¹ was calculated for $P_{ref} \leq 2.5$ mm h⁻¹ (Fig. 6 A2). The R²-value was 0.80 and 0.54, respectively. Measurements by the AS (SE) had a bias of -0.23 mm h⁻¹, an SDD of 0.57 mm h⁻¹ and a R²-value of 0.79

(Fig. 6 B1). For intensities classified as "slight", the sensor measured with a bias of -0.18 mm h⁻¹, an SDD of 0.37 mm h⁻¹ and a R²-value of 0.62 (Fig. 6 B2). The measurements from the AS in DD had a bias of 0.07 mm h⁻¹ and an SDD of 0.66 mm h⁻¹ with R² being 0.85 (Fig. 6 C1). Slight precipitation was measured with a bias of -0.07 mm h⁻¹, an SDD of 0.42 mm h⁻¹ and a R²-value of 0.60 (Fig. 6 C2).

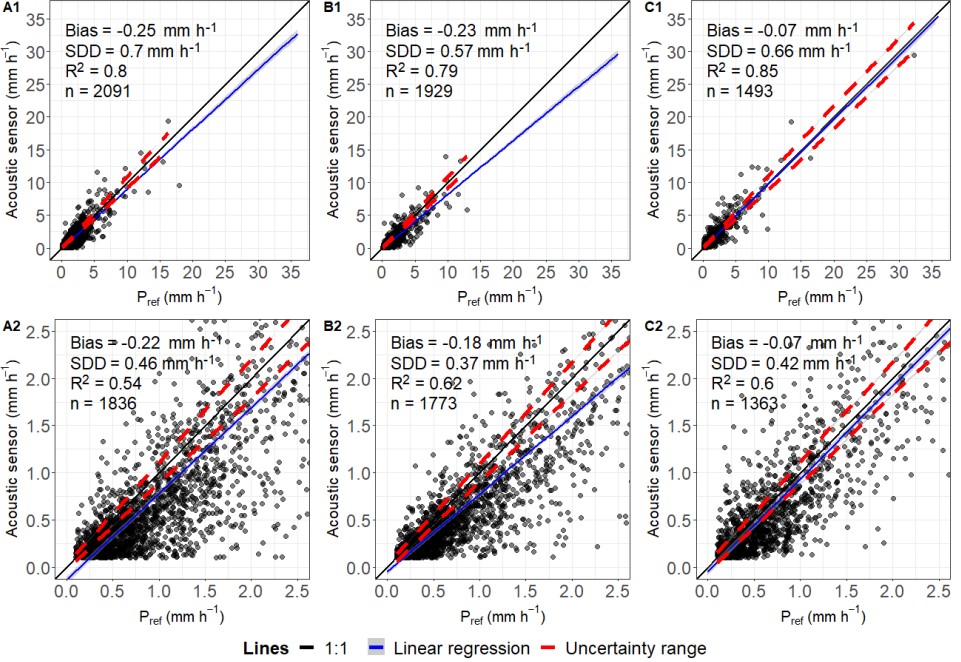


**Figure 6:** Comparison of hourly data determined by acoustic sensors and lysimeters ($P_{ref}$) and classified as "Rain". Plots A1–C1 include all measurements taken with both $P_{gauge}$ and $P_{ref}$ are ≥ 0.1 mm h⁻¹. Plots A2–C2 show slight precipitation events and key values are calculated for $P_{ref}$ between 0.1–2.5 mm h⁻¹. A1, A2: Rollesbroich lysimeter station. B1, B2: Selhausen lysimeter station. C1, C2: Dedelow lysimeter station.

The LD (RO) tended to overestimate the precipitation with a general bias of 0.01 mm h⁻¹ and an SDD of 0.37 mm h⁻¹ (Fig. 7 A1). The R²-value of 0.95 suggests a good correlation with the reference. For slight precipitation events, the bias was 0.00 mm h⁻¹ with the SDD being 0.18 mm h⁻¹ and R² 0.90 (Fig. 7 A2).

The LD (SE) showed biases of 0.02 mm h⁻¹ for all measurements and 0.04 mm h⁻¹ for $P_{ref} < 2.50$ mm h⁻¹ (Fig. 7 B1 and B2). The SDDs were 0.61 and 0.43 mm h⁻¹ respectively. The R²-values (0.75 and 0.58) were lower compared to the ones from the





other disdrometer at RO. The number of recorded hours with precipitation occurring at both sides was higher for the plots with the disdrometers compared to the plots of the other gauges due to the high measuring resolution of the disdrometer.

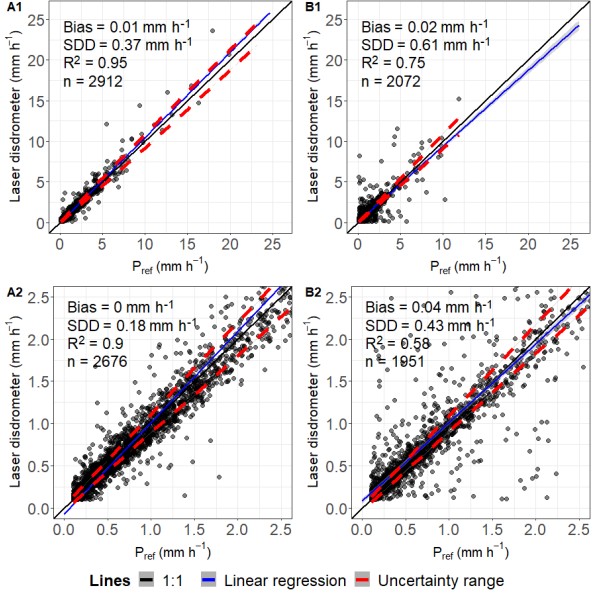

**Figure 7**: Comparison of hourly data determined by laser disdrometer and lysimeters ($P_{ref}$) and classified as "Rain". Plots A1–B1 include all measurements taken with both $P_{gauge}$ and $P_{ref}$ being $\geq 0.1$ mm h$^{-1}$. Plots A2–B2 show slight precipitation events and key values are calculated
for $P_{ref}$ between 0.1–2.5 mm h$^{-1}$. A1, A2: Rollesbroich lysimeter station. B1, B2: Selhausen lysimeter station.

### 3.4 Influence of wind speed

The CRs from TB1 (RO) and TB (SE) as functions of the wind speed at gauge height were mostly in the range of 30–100 % and did not in- or decrease with increasing wind speed at gauge height (Fig. 8 A and C). A major influence of the wind on hourly precipitation data therefore could not be identified. The CRs from TB2 (RO) showed a tendency to decrease at wind
speeds greater than 5 m s$^{-1}$ (Fig. 8 B). In contrast, the CRs from TB (DD) tended to be above 100 % after the same threshold and the regression line implies a positive correlation with the wind speed (Fig. 8 D). The CRs from the WGs did not exhibit any correlation between the hourly precipitation data and the wind speed (Fig. 9 A to C), even with the separation according to the availability of the wind shield.

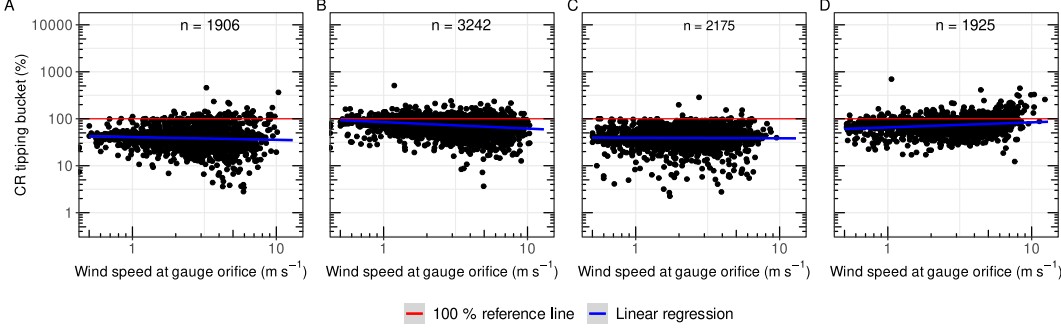

**Figure 8:** Catching ratios (CRs) of the tipping buckets as functions of the wind speed at gauge height for all sites, $P_{ref} \geq 0.1$ mm h$^{-1}$ and precipitation classified as "Rain". A: Rollesbroich lysimeter station. B: Rollesbroich EC station. C: Selhausen lysimeter station. D: Dedelow lysimeter station.



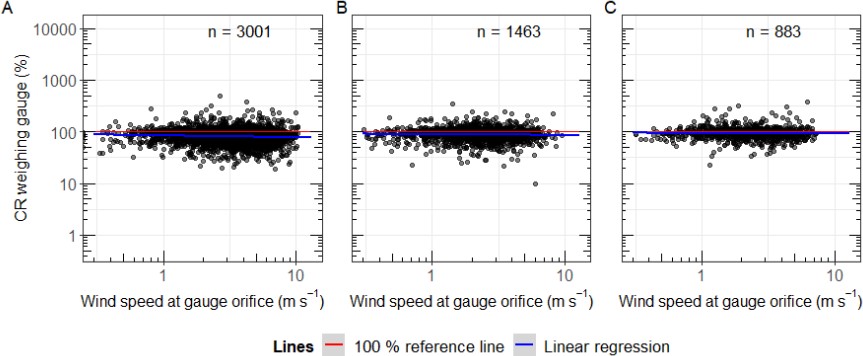

**Figure 9:** Catching ratios (CRs) of the weighable gauges as functions of the wind speed at gauge height, $P_{ref} \geq 0.1$ mm h$^{-1}$ and precipitation classified as "Rain". A: Rollesbroich lysimeter station. B: Selhausen lysimeter station, before wind shield was installed. C: Selhausen lysimeter station, after wind shield was installed.

A slight increase of CRs at higher wind speeds could be traced back to an influence of the wind speed for the ASs at all sites (Fig. 10 A to C). The CRs of the LDs generally did not indicate dependencies on the wind speed at gauge height (Fig. 11 A to B). The CRs of the LD in SE generally spread to a greater extend around the 100 % reference line compared to LD in RO.

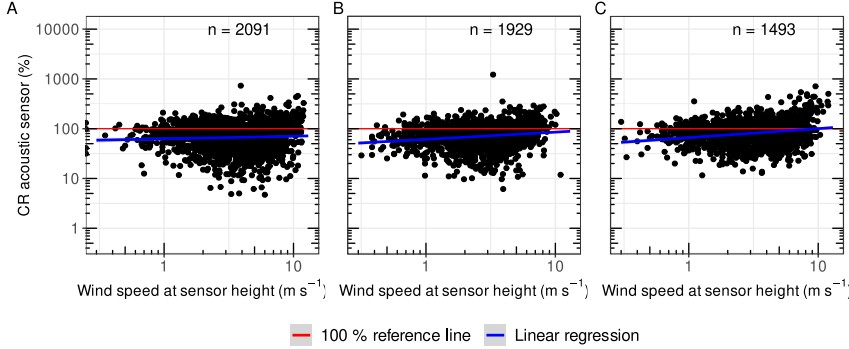

**Figure 10.** Catching ratios (CRs) of the acoustic sensors as functions of the wind speed at gauge height for all sites, $P_{ref} \geq 0.1$ mm h$^{-1}$ and precipitation classified as "Rain". A: Rollesbroich lysimeter station. B: Selhausen lysimeter station. C: Dedelow lysimeter station.

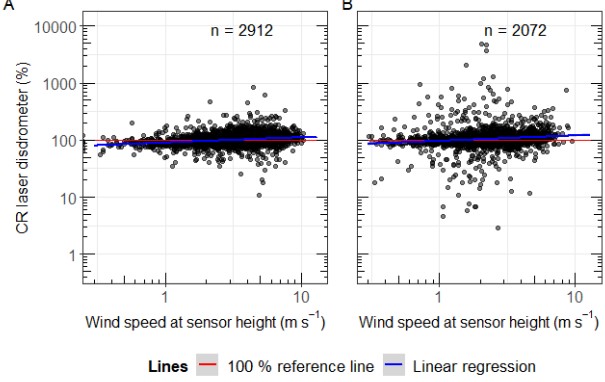

**Figure 11.** Catching ratios (CRs) of the laser disdrometers as functions of the wind speed at gauge height for all sites, $P_{ref} \geq 0.1$ mm h$^{-1}$ and precipitation classified as "Rain". A: Rollesbroich lysimeter station. B: Selhausen lysimeter station.





### 3.5 Precipitation correction

Applying the dynamic correction model (DCM) and the approach derived from Richter (1995) reduced the bias of all corrected measurements for the TBs and WGs (Table 6). Generally, the DCM resulted in a greater bias reduction than the other method. Both correction methods had no systematic effect on the SDD and R²-values, but the CRs have been increased by 14 and 9 %

for the TBs and by 10 and 11% for the WGs due to the application of the DCM and method after Richter, respectively. The data correction also led to an increased number of hours exceeding the threshold of 0.1 mm h$^{-1}$ for the WGs (Table 6 and 7). For the TBs, the aggregation of hourly to daily precipitation data during a processing step of the method derived from Richter resulted in a reduction of observations.

**Table 6:** Statistics of the corrected precipitation data classified as "Rain" for tipping bucket (TB) and weighable gauges (WG) at sites in
Rollesbroich (RO), Selhausen (SE) and Dedelow (DD). Due to the aggregation of hourly values and therefore the number of days with incomplete data as well as the threshold of 0.1 mm, which is met only after the correction, the number of observations (n) differed depending on the correction method. "Intercept" and "Slope" show the intercept and slope of the linear regression of $P_{gauge}$ and $P_{ref}$.

| Site | Device | Cor. method | Intercept | Slope | Bias | SDD | R² | n |
|---|---|---|---|---|---|---|---|---|
| | | | (mm h$^{-1}$) | (-) | (mm h$^{-1}$) | (mm h$^{-1}$) | (-) | (-) |
| | TB1 | - | -0.08 | 0.50 | -0.69 | 0.90 | 0.83 | 1906 |
| | TB1 | DCM | -0.01 | 0.47 | -0.56 | 0.88 | 0.82 | 1906 |
| **Rollesbroich** | TB1 | Richter | 0.00 | 0.51 | -0.61 | 0.90 | 0.83 | 1831 |
| | TB2 | - | -0.11 | 0.93 | -0.18 | 0.31 | 0.95 | 3242 |
| | TB2 | DCM | 0.01 | 0.96 | -0.03 | 0.30 | 0.95 | 3242 |
| | TB2 | Richter | -0.03 | 0.94 | -0.09 | 0.30 | 0.95 | 3224 |
| | TB | - | -0.04 | 0.33 | -0.61 | 0.85 | 0.79 | 2110 |
| **Selhausen** | TB | DCM | 0.13 | 0.35 | -0.51 | 0.83 | 0.78 | 2110 |
| | TB | Richter | 0.11 | 0.34 | -0.54 | 0.85 | 0.78 | 2019 |
| | TB | - | -0.01 | 0.79 | -0.20 | 0.41 | 0.95 | 1925 |
| **Dedelow** | TB | DCM | 0.11 | 0.81 | -0.06 | 0.42 | 0.93 | 1925 |
| | TB | Richter | 0.07 | 0.80 | -0.11 | 0.41 | 0.94 | 1925 |
| | WG | - | -0.06 | 0.91 | -0.15 | 0.24 | 0.98 | 3001 |
| **Rollesbroich** | WG | DCM | 0.01 | 0.95 | -0.04 | 0.22 | 0.98 | 3107 |
| | WG | Richter | 0.01 | 0.93 | -0.06 | 0.23 | 0.98 | 3185 |
| | WG | - | -0.02 | 0.95 | -0.07 | 0.17 | 0.98 | 2346 |
| **Selhausen** | WG | DCM | 0.01 | 0.99 | 0.01 | 0.17 | 0.98 | 2439 |
| | WG | Richter | 0.05 | 0.97 | 0.03 | 0.16 | 0.98 | 2497 |

**Table 7:** Catching ratios (CRs) of corrected tipping bucket (TB) and weighable gauge (WG) data (DCM = Dynamic Correction Model;
RI = correction based on the method derived from Richter, 1995). Only numbers (n) of hours are considered, when all devices at a site were active and at least one device (TB, WG, AS) did measure precipitation (P) classified as "Rain" of at least 0.1 mm h$^{-1}$. Latter conditions are differently met after the correction with the two correction methods, resulting in divergent numbers of observations and associated $P_{ref}$.

| Site | Year | n DCM | n RI | $P_{ref}$ DCM | $P_{ref}$ RI | $CR_{TB1}$ DCM | $CR_{TB1}$ RI | $CR_{TB2}$ DCM | $CR_{TB2}$ RI | $CR_{WG}$ DCM | $CR_{WG}$ RI |
|---|---|---|---|---|---|---|---|---|---|---|---|
| | (-) | (h) | (h) | (mm) | (mm) | (%) | (%) | (%) | (%) | (%) | (%) |
| | 2015 | 799 | 801 | 710 | 693 | 57.8 | 53.0 | 99.9 | 92.2 | 97.4 | 94.0 |
| **Rollesbroich** | 2016 | 1009 | 1054 | 803 | 805 | 29.5 | 26.9 | 95.0 | 88.7 | 92.4 | 92.0 |
| | 2017 | 1027 | 1020 | 882 | 856 | 39.6 | 36.5 | 98.3 | 93.0 | 94.7 | 94.0 |
| | 2018 | 709 | 705 | 584 | 557 | 41.3 | 38.0 | 96.3 | 90.8 | 99.7 | 99.1 |
| | 2015 | 760 | 744 | 601.36 | 552.4 | 52.5 | 48.9 | | | 100.5 | 103.7 |
| **Selhausen** | 2016 | 721 | 733 | 556.31 | 529 | 51.4 | 48.1 | | | 97.3 | 100.8 |
| | 2017 | 598 | 616 | 432.88 | 411.6 | 51.3 | 48.0 | | | 99.6 | 105.5 |
| | 2018 | 570 | 611 | 389.81 | 391.6 | 28.8 | 25.8 | | | 104.5 | 107.5 |





| | | | | | | |
|---|---|---|---|---|---|---|
| **Dedelow** | 2015 | 571 | 571 | 407.52 | 407.5 | 103.3 | 97.2 |
| | 2016 | 641 | 641 | 410.68 | 410.7 | 97.2 | 91.8 |
| | 2017 | 719 | 719 | 682.1 | 682.1 | 87.5 | 83.1 |
| | 2018 | 407 | 407 | 271.31 | 271.3 | 106.7 | 100.9 |

## 4 Discussion

### 4.1 Lysimeter measurements, reference values and the influence of non-rainfall events

The uncertainty ranges of the reference values and the overall strong correlation between the intensity measurements of the lysimeters within a site (Appendix B) demonstrate, that under field conditions and different climates, the lysimeter measurements were in good agreement with each other over the entire observation period at all sites. The uncertainties of the reference values meet the requirements defined by the WMO (2018) for gauges used in the field. They decreased with increasing precipitation intensity, reaching a threshold of 5 % at about 1 to 2 mm h$^{-1}$, which was observed at all study sites and

regardless of the vegetation cover of the lysimeters. Additionally, lysimeter data at the sites showed R²-values of 1 with a maximum bias of 0.01 mm h$^{-1}$ and a maximum SDD of 0.05 mm h$^{-1}$ (Appendix B1 to B3), indicating a good correlation. Since lysimeter measurement uncertainties increase exponentially with decreasing precipitation intensity from $P_{ref} < 1$ mm h$^{-1}$, calculating an average intensity from multiple lysimeters seems necessary for these data. For long-term periods, uncertainties were probably less important because of the assumed normally distributed deviations from the average.

In total, when comparing the cumulated precipitation within a certain period, the lysimeters registered more precipitation than any other precipitation gauge. It could therefore be assumed that the lysimeter measurements and therefore the reference values are closer to the "true" precipitation than any of the other measuring method compared. However, comparing total precipitation amounts over long-term periods between the gauges and lysimeters, even if filtered carefully, might be biased because of non-rainfall events (NRE), which contributed to the recorded precipitation measured by the lysimeters. NREs which occurred

before, during or after a regular precipitation event within respective time interval were hence classified as precipitation, while other precipitation gauges did not record these. NRE could thus lead to overestimations of lysimeter precipitation intensities, although the average intensity of a NRE is small compared to the precipitation intensity (RO: 0.012 mm h$^{-1}$; SE: 0.009 mm h$^{-1}$; DD: 0.013 mm h$^{-1}$) and the distribution of NRE intensities (Fig. 12 A to C) indicated that these presumably did not heavily bias hours with regular precipitation. Applying these averages for NRE intensities on all hours which were investigated for the comparison of precipitation totals (Table 6), only 0.7 % (RO), 0.6 % (SE) and 0.8 % (DD) of the total registered precipitation

would have been attributed to NRE. This indicates a possibly small impact of NREs on the results of this study. However, NRE (i.e., dew, hoar frost and fog) can contribute a substantial amount of water for the ecosystem at the annual scale (Forstner et al., 2021; Groh et al., 2019), which demonstrates clearly that high-precision lysimeters are ideal tools measuring the different NRE of ecosystems. A higher temporal resolution of the data, possibly 10-min intervals, could help to temporally delineate

precipitation events and NRE, as used by Groh et al. (2018b).





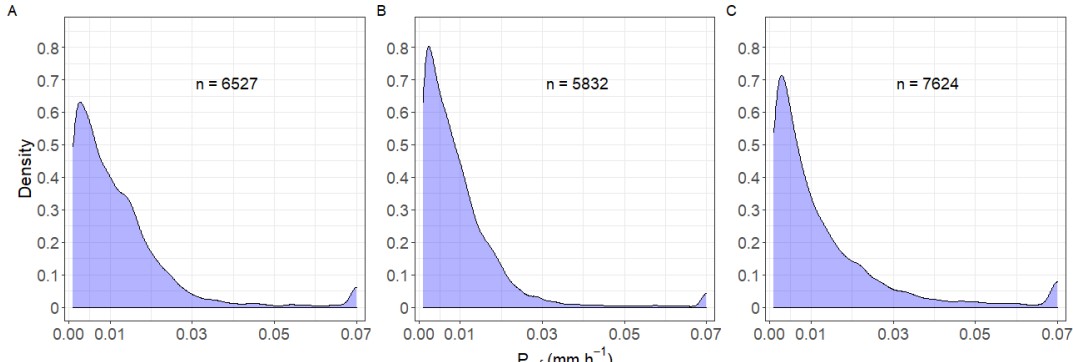

**Figure 12:** Distribution of water amounts formed during non-rainfall events ($P_{ref} > 0$ mm h$^{-1}$; $P_{gauge} = 0$ mm h$^{-1}$) occurring between sunset and sunrise. Hourly values of non-rainfall events were limited to a maximum value of 0.07 mm h$^{-1}$ according to Monteith and Unsworth (2013). A: Rollesbroich lysimeter station. B: Selhausen lysimeter station. C: Dedelow lysimeter station.

### 4.2 Evaluation of gauge types for the comparison of precipitation data

The systematic deviations between the reference values and the precipitation intensity measurements by TB1 in RO (Fig. 4 A) and the TB in SE (Fig. 4 C) on one side and the better fitting data from TB2 (RO; Fig 5 B) and TB in DD (Fig. 4 D) on the other side, can only be explained by divergent calibrations. All TBs were of the same type and only differed in the equipped heating module, which did not affect the measurements compared, since it only worked at low temperatures and was intended to melt solid precipitation. The TB (DD) also had a greater catching surface area of 400 cm² compared to 200 cm² at other sites. However, this effect should be deducted by the internal processing. A spatial influence on the measurements was also unlikely since TB2 in RO was installed 30 m away from the lysimeter station and measured values closer to the reference than TB1, which was located only 5 m beside the lysimeters. The maintenance and service intervals were nearly equal at all sites. No or incorrect calibration could lead to a systematic underestimation of precipitation measurements, especially regarding TBs (Calder and Kidd, 1978; Niemczynowicz, 1986; Shedekar et al., 2016). If the precipitation amount of the internal processing software does not agree with the amount of liquid, which is poured by a tip of the bucket, the bias increases with the amounts of tips. Thus, an increased underestimation of the precipitation intensity can be recognized with increasing reference values. Kohfahl and Saaltink (2020) also found that TBs located at the same site differed in the amount of precipitation measured during six rain events compared to a high-precision weighable lysimeter for bare soil conditions. The authors assumed that the TB, which showed significant errors with minor amounts of rainfall, was affected by an individual technical problem. However, also the TB2 (RO; Fig. 4 D) showed a systematic underestimation of precipitation intensities, in particular for slight precipitation intensities. The TB (DD) was also prone to this circumstance and showed in addition outlier up to 9 mm h$^{-1}$ at slight precipitation events. This might also be fixed with a dynamic calibration, although the occurrence of outliers towards an overestimation was exceptional compared to the other TBs. Considering that the hourly deviations led to underestimations of up to 67.7 % compared to the reference (TB RO; n = 3483 h), calibrations should be conducted regularly when operating a TB. TBs were also subject to the wetting loss, since precipitation can adhere on the gauge's inlet due to rough material surfaces and thus can be evaporated or sublimated without being measured (Sevruk, 1974). Due to the gauge´s specific resolution, which is limited for TB gauges by the volume of a bucket, certain amounts of precipitation could not be registered and therefore got lost (loss of trace precipitation; Seibert and Morén, 1999; Sugiura et al., 2003; Yang et al., 1999b). This is particularly relevant because a minimum threshold of 0.1 mm h$^{-1}$ can be set for the overall comparison, but precipitation amounts below or above the gauge's resolution were not recorded by the respective TBs, but only by the lysimeters. This could lead to an underestimation of precipitation amounts compared to the reference. In addition to other external influences such as wind and temperature, this is reflected in the slight, systematic underestimation of the better performing TBs.





The WGs had the best correlation with low biases and small deviations in the precipitation totals, when the internally
unprocessed data was compared to the reference. At both investigation sites, the devices underestimated the precipitation
intensity consistently but within or close to the uncertainty range of the reference values. The weighable gauge (WG; RO) had
a slightly lower R²-value and SDD, which might go back to generally higher wind speeds at the site. The installation of the
wind shield had a positive effect on reducing the bias and SDD of the WG (SE; Fig. 5). This results agreed well with a previous
study which showed that Alter wind shield on a WG (Pluvio²) reduced the bias and increased the accuracy compared to an
unshielded WG (Kochendorfer et al., 2017a). Overall, the WG also achieved the highest CR in the comparison of absolute
precipitation at both sites. Here, too, there was a difference of 14.7–8.0 % compared to the reference, which could be explained
by the gauge's resolution and the wind effect. These results generally support approaches from other studies (e.g., Fehlmann
et al., 2020; Johannsen et al., 2020) that used the WG as a reference device to study other gauges. Although the deviations of
the WG data from the lysimeter data were still relatively high, this was the best correlation of all the investigated gauges in
this study.

The hourly precipitation intensities determined by the AS were inaccurate compared to the reference with both, over- and
underestimated intensities, shown by SDDs of up to 0.7 mm h$^{-1}$. External or internal data processing implementing an artificial
time lag would lead to similar broad scattering (Fig. 6). However the data were tested for known errors indicating a time lag
and no such influence could be identified. Salmi and Ikonen (2005) pointed out, that variations of the AS measurements were
more of a stochastic than systematic origin. Variation in the shape and velocity of the hydrometeors caused by air movements
were the main reason for erroneous measurements. Stochastic errors were also produced by the surface wetness and
construction of the sensor itself due to sensitivity variations over the sensor area. In a comparison of multiple instruments to
measure precipitation, Liu et al. (2013) found the lowest correlation coefficient and the largest SD while comparing their
reference, a TB, with the AS with a temporal resolution of one minute. Moreover, the AS overestimated the rainfall
accumulation and recorded little higher rain intensity compared to their reference, when the rain intensity was less than 20
mm h$^{-1}$ (Liu et al., 2013). According to Haselow et al. (2019), the AS showed overestimations by highly positive error values,
compared to lysimeter reference data on a daily basis. Even though the hourly deviations in precipitation intensity were partly
above the reference values, less absolute precipitation was measured by the ASs at all research sites than specified by the
reference values. However, there were major differences between the sites. The different ASs in RO and SE registered 69.2 %
and 73.2 % of the reference values, respectively, and in DD 86.8 %. The variation between the sites may exist due to the
selection of hours with precipitation recorded by either the TB, AS or WG. The latter was absent in DD, so the threshold of
0.1 mm h$^{-1}$ was more often reached by the AS at this site compared to the other sites. Therefore, precipitation registered only
by the WG is not affecting the CRs of the ASs at DD.

Precipitation measurements performed by the LD had a small bias compared to the reference, which was probably primarily
due to the resolution of the measuring device of 0.001 mm h$^{-1}$. The overall good correlations are reflected in the absolute values
measured over a longer period, which can be compared with the CRs of the WGs. The deviations from the reference could be
attributed to stochastic influences of drop size distributions and conversions during the processing of the measured data.
According to a study by Johannsen et al. (2020), recorded drop sizes, velocity distributions, and kinetic energy intensity
relationships were device-specific and showed similarities only for disdrometers of the same type across measurement sites.
Liu et al. (2013) found that small raindrops tend to be omitted in larger size raindrops due to shadow effects of light. Also, two
adjacent particles could appear as a large particle, resulting in wrong precipitation intensities (Lanzinger et al., 2006). The
wind direction could also influence measurements of optical instruments as well as splashing of large raindrops off the device
(Dengel et al., 2018). Hence an investigation of CRs as functions of the wind direction could reveal such interrelations.
Furthermore, the number of erroneous measurements was high compared to the other gauges. For the LD (SE) 534 hours must
be manually filtered, compared to 42 hours for the LD (RO). The data showed abrupt and isolated occurring hours with high
precipitation intensities (> 50 mm h$^{-1}$) as well as consecutive hours with varying precipitation intensities, which were clearly





decoupled from actual precipitation events. This could be explained by foreign objects interfering with the laser beam. Additionally, prolonged time periods occurred when the disdrometer continuously recorded fluctuating precipitation intensities with notable deviations from the reference. This error might be triggered by spider webs or insects intercepting the laser, or

water and dust particles on the sensor, which were known to cause such errors on optical disdrometer measurements (Adirosi et al., 2018; Heyn et al., 2018).

Ultimately, all the gauges studied tended to underestimate precipitation amounts compared to the reference, both on an hourly basis and over longer time periods. This means that calibrating large-scale weather simulations with such biased gauge data could lead to an underestimation of actual precipitation amounts in these models. The use of a less accurate precipitation data

set in environmental model calibration compromises parameter identification and reduces the ability of the model to simulate processes associated with water and solute transport in the critical zone (e.g., Groh et al., 2018a). For the calculation of small-scale water balances, biased precipitation data as an input variable could also distort the overall research results.

**4.3 Influence of wind speed on gauge precipitation measurements**

Of the two best performing TBs, one (TB2, RO) exhibited generally lower CRs with increasing wind speed (Fig. 8 B). This

finding agreed with the general assumption of the effects of wind-field formations on Hellmann-design gauges (Sevruk et al., 1989). The TB (DD) revealed increased CRs at higher wind speeds (Fig. 8 D). Usually, the contrary effect is assumed (Duchon and Biddle, 2010; La Barbera et al., 2002). A connection of this phenomenon to the precipitation intensity was not evident. The AS at the site also had a higher CR than the reference at high wind speeds (Fig. 10 D), but this pattern occurred also at other sites (Fig. 10 B to D). The TB (DD) is the only TB installed directly within the crop field. Thus, during crop season,

water from the surrounding vegetation might have dripped into the funnel and the crop might have functioned as wind protection. However, a comparison of data from the respective months with (April–July) and without available crops does not indicate such an effect.

The CRs of the WG (RO) did not indicate significant influences of wind speed on the precipitation data (Fig. 9) even though the splitting of the WGs (SE) dataset according to the availability of a wind shield revealed such influences affecting the

measured precipitation intensities of the WGs. The CRs for the AS data might imply a bias towards overestimation of precipitation intensity with increasing wind speeds. This phenomenon was recognized in the literature (Liu et al., 2013) and could be traced back to the gauges measuring principle. An wind induced, increased terminal velocity with which a hydrometeor hits the sensors surface, was directly converted to higher precipitation intensities (Salmi and Ikonen, 2005). Both LD tended to have increased CRs along with higher wind speeds, which might be related to the conversion from recorded drop

size distribution and vertical velocity of the raindrops to precipitation intensities.

**4.4 Evaluation of the precipitation data correction methods for TB and WG data**

To improve data quality with precipitation data correction during post-processing, the correction method must be selected and adapted to the measuring device collecting the data (WMO, 2018). Here, two different correction methods were applied to TB and WG data to examine their impact on the respective hourly precipitation data. Both approaches reduced the bias of the TB

data relative to the reference, which was a key goal of the precipitation data correction. The dynamic correction model (DCM) led to a generally greater reduction of the bias (TBs: -0.13 mm h$^{-1}$; WGs: -0.09 mm h$^{-1}$) for all gauges than the approach derived from the method of Richter (1995; TBs: -0.08 mm h$^{-1}$; WGs: -0.07 mm h$^{-1}$). As a result of the corrections, considerable amounts of precipitation have been added to the precipitation totals in the period under study. Especially the correction of data from TB2 (RO), WG (RO) and WG (SE) led to CRs of 97.4 (+15.7 %), 96.0 (+10.7 %) and 100.5 % (+8.5 %) compared to the

lysimeter reference. This indicates that the correction methods showed the best effect for the data with the initially highest quality, but possibly could lead to overestimations, which was found in overcorrection of precipitation data for WG (Se; Table 7).





Haselow et al. (2019) used the linear scaling method to reduce the bias of daily precipitation data from multiple rain gauges compared to lysimeter measurements. To do this, they applied the ratio between the monthly rainfall totals from lysimeters and rain gauges to the daily rainfall totals of the rain gauges. The method successfully reduced the bias of the daily precipitation data despite for periods of high precipitation intensities (Haselow et al., 2019). However, a distinction must be made between correction methods that correct precipitation data solely on the basis of reference data (e.g., Fang et al., 2015) and methods such as the DCM used in this study that correct individual physical and technical induced errors like wind field deformation and evaporation. For the latter approach, the availability of results from comparable studies with weighable lysimeter references is limited. For TB1 (RO) and TB (SE), the correction was not sufficient to compensate for the systematic biases from the reference, although the distortion probably came from calibration issues that were not intended to be corrected by the DCM. An overcorrection of the precipitation intensities for the WGs at high wind speeds cannot be ruled out, especially since the Alter wind shield has already reduced the influence of the wind on the WG (SE). Michelson (2004) found that the DCM resulted in more accurate precipitation estimates, although uncertainties in the treatment of measurements for some gauge types remain, which can be confirmed by the results of this work. The results also showed that the biases for all corrected TBs were reduced by applying the adjusted correction method according to Richter (1995), although high biases still remain for the TB1 (RO), TB (SE) and TB (DD). It must be taken into account that the orders of magnitude of these correction amounts were detached from individual environmental influences at the sites and only related to the manual Hellmann type gauge (Richter, 1995). Richter (1995) also stated that due to the large daily variability of the individual error-causing parameters, the statistical error of the calculations based on mean ratios was inevitably large and mainly in the magnitude of the correction amount. Therefore, the adjustment to correct the hourly measurements increased the statistical error. Based on given data availability of reference, gauge, and weather data, machine learning algorithms might be a promising tool to further optimise device-specific precipitation data corrections.

## 5 Conclusions

Precipitation data from three research sites of the TERENO-SOILCan network based on high-precision weighable lysimeters were used as reference to investigate the functionality and data quality of four different precipitation measurement methods. The arithmetic mean of the lysimeter measurements has proven to be an almost unbiased reference for the precipitation measurement method. The results of this study revealed that each gauge-based method (i.e., tipping bucket gauges, weighable gauges, acoustical sensors, and optical laser disdrometers) was affected differently by wind, precipitation intensity, measurement resolution and technical errors. All precipitation measurement methods underestimated the precipitation amounts for the observation period with deviations of 8 % to 67 % from the reference if only hours with precipitation intensities greater than 0.1 mm h$^{-1}$ were considered. This implies that point precipitation data should be treated as minimum values, especially when looking at cumulative totals over long time periods. When using hourly data for water balances or local projections of climate change, an uncertainty regarding over- and underestimation of point precipitation data must be taken into account, depending on the gauge type. The results confirmed that correction algorithms, which consider the influence of wind and other typical sources of error on the instruments, reduced the bias between reference and measurements and improved the catching ratios of hourly precipitation data from rain gauges (tipping bucket and weighable gauge) under different climatic conditions at three different TERENO-SOILCan test sites. The Dynamic Correction Model achieved higher average catch ratios compared to the correction approach derived from Richter. Therefore, only the Dynamic Correction Model might be the right tool to correct hourly precipitation data. Adequate reference data are crucial for testing and developing correction methods to overcome errors in precipitation measurements from standard point gauges. Observations from weighable, high-precision lysimeter (e.g., TERENO-SOILCan network) provide such data to improve estimates of point scale precipitation under different climatic conditions. Unbiased point-scale precipitation estimates are essential when estimating precipitation at larger





scales remotely from either ground-based weather radars or from satellites. Precipitation is the main driver of the hydrological
cycle and accurate data help to improve local weather and climate forecasts, which is particularly relevant in the context of
climate change.

**Appendix A**

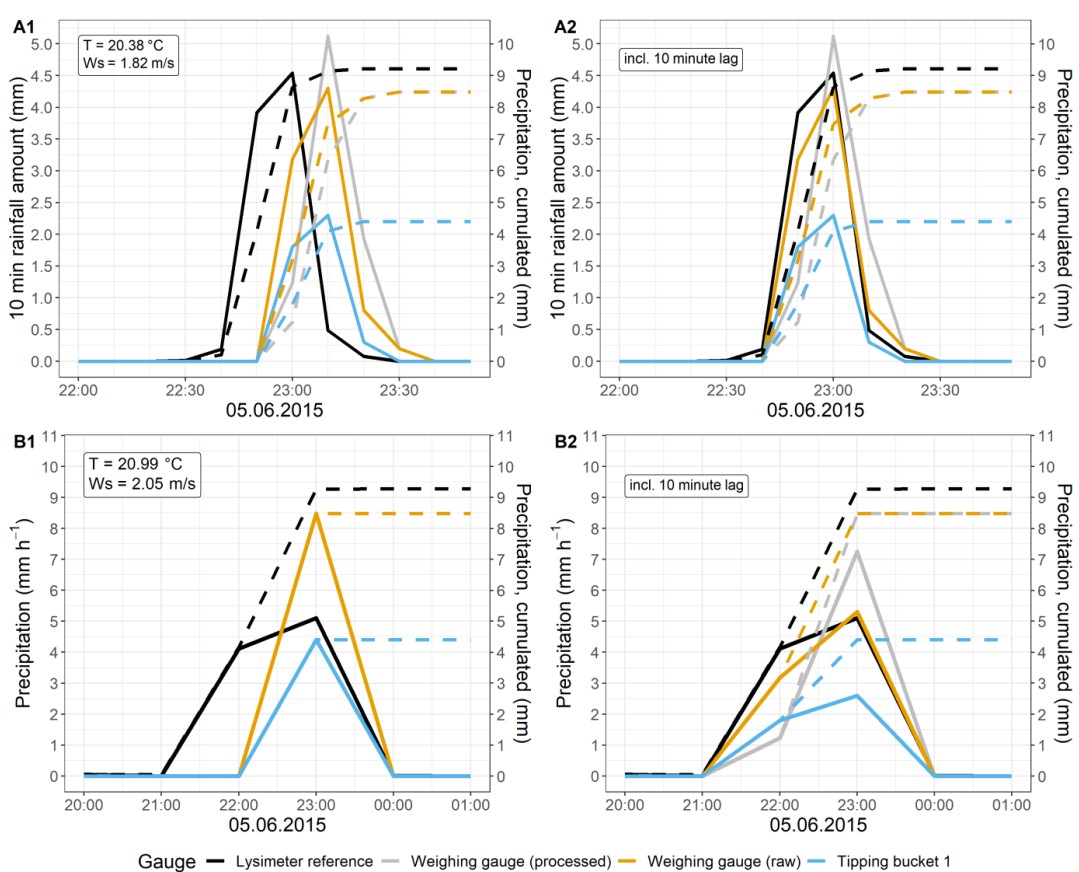

**Figure A1:** Time series of a precipitation event in Selhausen with a temporal resolution of A: 10 minutes, B: one hour. Solid lines display
the precipitation intensity (left y-axis) and dashed lines the cumulative precipitation (right y-axis). A1 and B1 show original data. A2 and
B2 show data with an implemented reverse time lag of 10 minutes for the tipping bucket and weighable gauge.





**Appendix B**

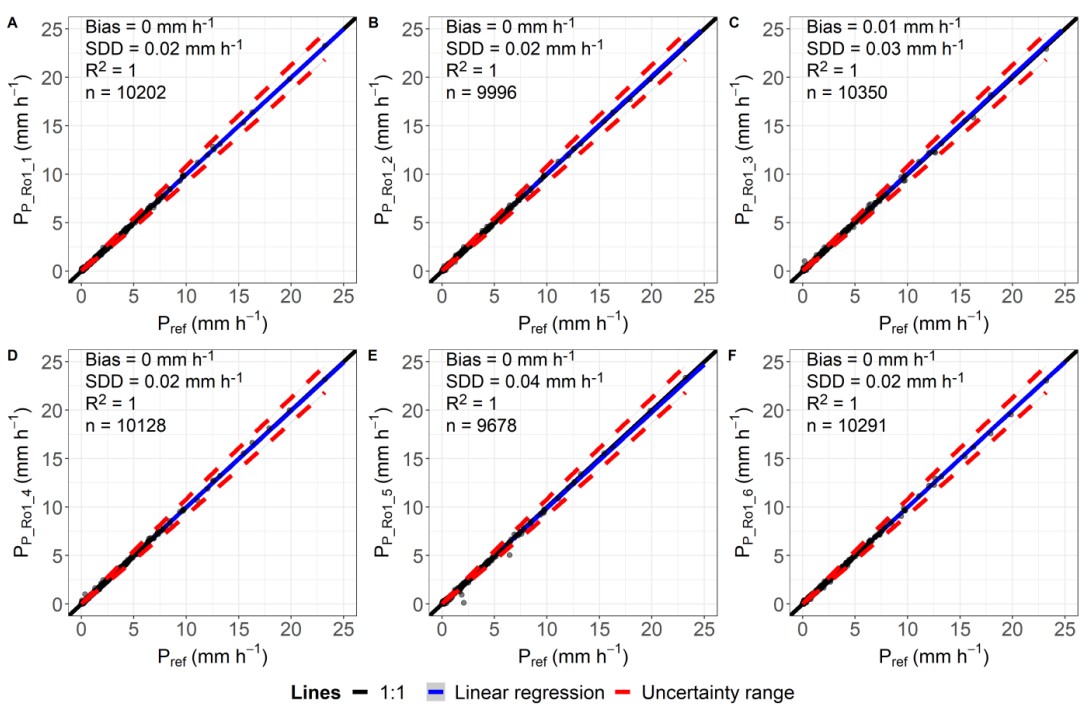

**Figure B1:** Comparison of hourly precipitation data classified as "Rain", determined by lysimeter and reference data ($P_{ref}$) for the lysimeter
station in Rollesbroich. Plots A–F include all hours of measurements taken where both $P_{lys}$ and $P_{ref}$ are $\geq 0.001$ mm h$^{-1}$. The vegetation grown
on the lysimeter is of a grassland type.

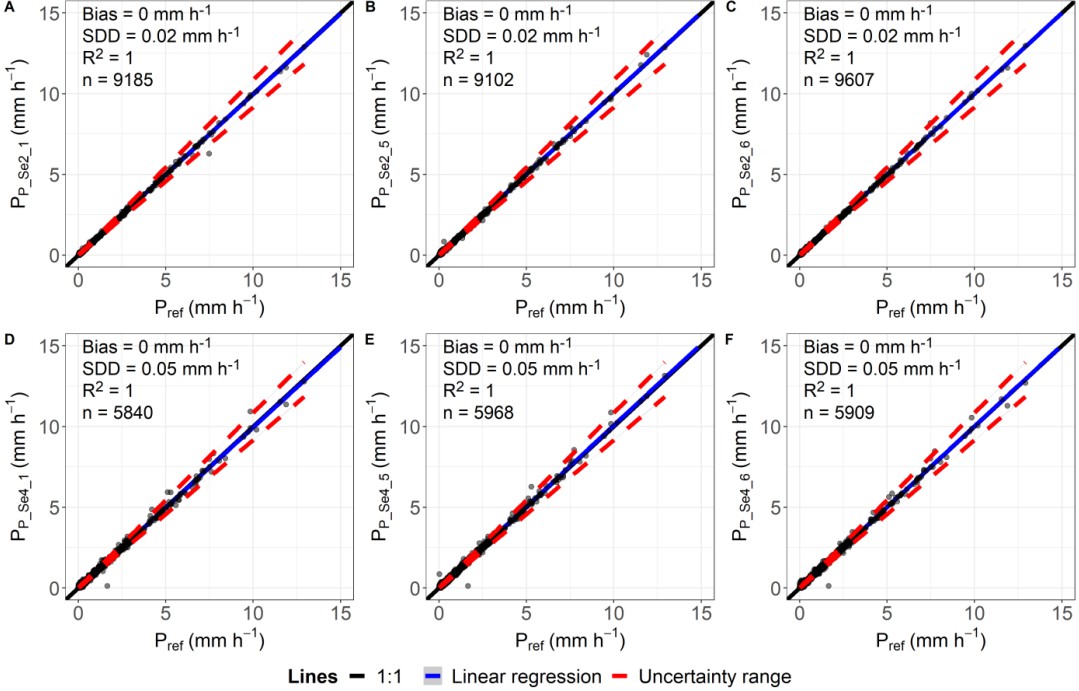

**Figure B2:** Comparison of hourly precipitation data classified as "Rain", determined by lysimeter and reference data ($P_{ref}$) for the lysimeter
station in Selhausen. Plots A–C include all hours of measurements taken where both $P_{lys}$ and $P_{ref}$ are $\geq 0.001$ mm h$^{-1}$ and the vegetation



grown on the lysimeter is of a grassland type. Plots D–F include all hours of measurements taken where both $P_{lys}$ and $P_{ref}$ are $\geq 0.001$ mm h$^{-1}$ and the vegetation grown on the lysimeter is of an arable land type.

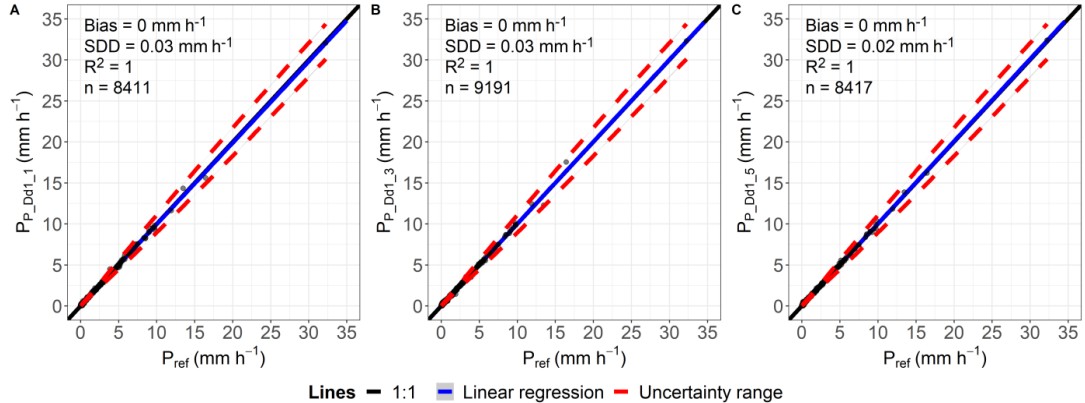

**Figure B3:** Comparison of hourly precipitation data classified as "Rain", determined by lysimeter and reference data ($P_{ref}$) for the lysimeter station in Dedelow. Plots A–C include all hours of measurements taken where both $P_{lys}$ and $P_{ref}$ are $\geq 0.001$ mm h$^{-1}$. The vegetation grown 715 on the lysimeter is of an arable land type.

### Data availability

All raw data for the specific lysimeters, precipitation gauges and weather stations from Rollesbroich and Selhausen can be freely obtained from the TERENO data portal (https://teodoor.icg.kfa-juelich.de/ddp/index.jsp (last access: 20 October 2022; Kunkel et al., 2013) with respective ID codes.

Rollesbroich lysimeter station: RO_BKY_010 (gauge and weather data), RO_Y_01 (lysimeter data).

Rollesbroich Eddy covariance station: RO_EC_001 (gauge data).

Selhausen lysimeter station: SE_BDK_002 (gauge and weather data), SE_Y_02 (lysimeter: Se_Y_021, Se_Y_025, Se_Y_026), SE_Y_04 (lysimeter: Se_Y_041, Se_Y_045, Se_Y_046).

Dedelow lysimeter station: Dd_K_01 (gauge and weather data), Dd_Y_01 (lysimeter data). The data for the experimental 725 station in Dedelow can be acquired upon request from Jannis Groh.

### Author contributions

TP conceived the experiments. JG, TP, and TS had the idea and designed the study. JG provided the data for the corresponding lysimeter stations. HHG and BR guided the conceptualization and the internal review process. TS performed the data analysis and wrote the paper with equal contributions from all co-authors.

### 730 Competing interests

The authors declare that they have no conflict of interest.

### Acknowledgements

We acknowledge the support of TERENO and SOILCan, which were funded by the Helmholtz Association (HGF) and the Federal Ministry of Education and Research (BMBF). We thank the colleagues at the corresponding lysimeter station for their 735 kind support: Jörg Haase and Gernot Verch (Dedelow); and Werner Küpper, Ferdinand Engels, Philipp Meulendick, Rainer Harms and Leander Fürst (Rollesbroich and Selhausen).





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
