# Peer review of "Evaluation of precipitation measurement methods using data from precision lysimeter network"

_Hydrology and Earth System Sciences, 2022_

## Author Comment (AC2)

**Supplement for reply on "Comment on hess-2022-370', Anonymous Referee #2, 07 Feb 2023"**

[Figure]

Figure 1. Comparison of hourly precipitation data determined by a weighable gauge and lysimeters ($P_{ref}$) from Rollesbroich lysimeter station.

[Figure]

Figure 2. Comparison of hourly precipitation data determined by an acoustic sensor and lysimeters ($P_{ref}$) from Rollesbroich lysimeter station.

[Figure]

Figure 3. Catching ratios (CRs) of the acoustic sensor (Selhausen lysimeter station) as functions of the wind speed at sensor height and precipitation classified as "Rain". The data was categorised and averaged with each category containing data within a range of 0.1 m s$^{-1}$ for wind speeds ranging from 0.0 to 12.0 m s$^{-1}$. A: Data on 10-min basis with $P_{gauge}$ and $P_{ref} \geq 0.1$ mm 10min$^{-1}$; B: Data on hourly basis with $P_{gauge}$ and $P_{ref} \geq 0.1$ mm h$^{-1}$.

[Figure]

Figure 4. Catching ratios (CRs) of the tipping bucket gauge (Rollesbroich EC station) as functions of the wind speed at gauge height. $P_{gauge}$ and $P_{ref} \geq 0.1$ mm h$^{-1}$ and precipitation classified as "Rain". The data was categorised and averaged with each category containing data within a range of 0.1 m s$^{-1}$ for wind speeds ranging from 0.0 to 12.0 m s$^{-1}$.

---

## Author Response (AR1)

In the following we respond to the referee comments after we revised the manuscript. The revised manuscript is made available with and without detailed comments on the revisions. All responses in this document which include line numbers refer to the revised manuscript version comprising track changes and comments.

**Response to comments by Referee #1**

"The paper by Tobias Schnepper et al. focuses on evaluating several precipitation measurement techniques based on data from the TERrestial Environmental Observatories (TERENO)-SOILCan lysimeter network as a reference. The topic is important, and the paper is well-structured and written. I have, however, several concerns that require clarification and/or revisions, as described below:"

We thank anonymous Referee #1 for the constructive feedback and the concerns raised.

1) Reliability of precipitation measurements from the lysimeters: The assumption of no ET during precipitation interval and vice versa seems valid only at very short time intervals. Within an hour interval, I assume precipitation and ET can co-occur, especially for very wet soil and high evaporative demands. I think the authors should support this assumption, which underlies the precipitation computation. Otherwise, the reliability of the reference data is questionable.

We agree that within an hour, $ET_a$ and atmospheric water input (AWI; comprising precipitation and non-rainfall water) can occur in relevant amounts. This could lead to underestimations of either $ET_a$ or AWI when using hourly weight data. In our investigation, we solved this issue by using minutely resolved lysimeter weight data. Here, each minute is assigned either $ET_a$ or AWI, based on positive or negative lysimeter mass change. Although this still does lead to general underestimations in both, the AWI and $ET_a$ data, since these processes are also affecting minutely data, the dominating process can be identified more consistently with minutely resolution. We adjusted the respective passage (line 240 – 243) as followed: "Lysimeter data, which are available as time series of mass changes in minute resolution, were pre-processed in three four steps: i) check of the minutely monitored raw data time series' for plausibility (manually and automatically), ii) application of the AWAT-filter to reduce the impact of noise on the raw data of the lysimeter mass changes (Peters et al., 2017), iii) classifying $ET_a$ and AWI data according to Eq. 1-4 assuming that only $ET_a$ or AWI can occur within the respective 1-minute time step, and iv) summing the minutely to hourly values; note that hours with missing values were marked as not available (NA). "

2) Eq. 2: I assume P here includes NRE? It is unclear why NRE is presented in Eq. 1 but ignored until section 2.5.1. How come? Please clarify this.

We introduced the term AWI (line 178 – 188) to differentiate between precipitation and non-rainfall water (NRW) and to improve the accuracy of our formulations throughout the whole manuscript. Also, we replaced the term non-rainfall event (NRE) by NRW. The process to determine AWI and distinguish it from $ET_a$ is technically identical for both precipitation and NRW, involving weight measurement, filter application, and hourly data aggregation. The lysimeter reference calculation is also performed similarly for both types of water. However, in the gauge comparison of precipitation data, only the values above 0.1 mm $h^{-1}$ are considered as reference data. Therefore, it is assumed that precipitation is the dominant process responsible for the weight increase. Thus, we used the term "$P_{ref}$" to refer to this result.

We adjusted the passage from line 240 – 243 (see answer 1) and added a respective passage in lines 261 – 253: "The procedure is conducted for precipitation and NRW simultaneously; the following equations refer to the calculation for the reference precipitation, for which we only use AWI above 0.1 mm $h^{-1}$, which is assumed to be dominantly contributed by precipitation. The identification of non-rainfall water will be targeted in chapter 2.5.1."

3) More on NRE: according to former studies, NRE is a primary source of error for standard gauges compared to lysimeter precipitation data. It is, therefore, crucial this component would be as accurate as possible. However, the rules to determine NRE cases are very "ad-hoc" and probably specific to a given site. Furthermore, is it reasonable to assume that within an hour interval, precipitation will be either NRE or not? Isn't it possible to have rainfall and NRE together within the hour?

The general assumption to identify NRW is that all measurement methods except the disdrometer (fog) cannot detect these. NRW predominantly occur between sunset and sunrise (e.g. Monteith and Unsworth, 2013). We selected the respective times so that the time between sunset and sunrise is over rather than underestimated. In order to exclude erroneous data, in which case the measurement methods did not detect precipitation, but the lysimeters did, we limited the maximum value of AWI to 0.07 mm h$^{-1}$. The data showed that only 2.33, 1.15 and 3.18 % of the hours with NRW would have exceed the threshold of 0.07 mm h$^{-1}$ in Rollesbroich, Selhausen and Dedelow, respectively. We expect that we could identify NRW accurately enough to describe these data with statistics. We included these in a table in the Appendix (E1) and found that during the investigation period, around 17.3, 14.3 and 20.7 mm a$^{-1}$ of NRW have been deposited at RO, SE and DD, respectively (Table 1). This is equal to 2.3, 2.9 and 4.6 % of the average annual precipitation at the sites. Applying the mean of NRW on all hours which were investigated for the comparison of precipitation totals, only 1.3 % (RO), 1.2 % (SE) and 1.6 % (DD) of the total registered precipitation would have been attributed to NRW. This indicates that the systematic influence of NRW on the precipitation amounts, if they occur during the same hour, is limited.

**Table 1:** Statistics on non-rainfall water (NRW) recorded by the lysimeter within the investigated period. NRW was identified, when no gauge but the lysimeter or disdrometer recorded atmospheric water input (AWI) between sunrise and sunset. The statistics are given for NRW ≤ 0.07 mm h$^{-1}$.

| | Year | Sum | Median | Mean | Standard deviation | n | n incl. $P_{ref} > 0.07$ mm h$^{-1}$ |
|---|---|---|---|---|---|---|---|
| | (-) | (mm h$^{-1}$) | (mm h$^{-1}$) | (mm h$^{-1}$) | (mm h$^{-1}$) | (h) | (h) |
| Rollesbroich | 2015 | 14.3 | 0.010 | 0.012 | 0.010 | 1245 | 1263 |
| | 2016 | 20.1 | 0.008 | 0.011 | 0.011 | 1807 | 1860 |
| | 2017 | 18.7 | 0.009 | 0.012 | 0.011 | 1585 | 1631 |
| | 2018 | 16.1 | 0.007 | 0.010 | 0.009 | 1639 | 1668 |
| Selhausen | 2015 | 15.8 | 0.008 | 0.011 | 0.010 | 1505 | 1527 |
| | 2016 | 14.1 | 0.006 | 0.009 | 0.082 | 1636 | 1652 |
| | 2017 | 14.3 | 0.007 | 0.009 | 0.086 | 1544 | 1563 |
| | 2018 | 12.9 | 0.006 | 0.008 | 0.007 | 1636 | 1652 |
| Dedelow | 2015 | 22.0 | 0.009 | 0.012 | 0.011 | 1767 | 1807 |
| | 2016 | 24.0 | 0.007 | 0.012 | 0.013 | 1944 | 2007 |
| | 2017 | 20.4 | 0.009 | 0.013 | 0.013 | 1532 | 1593 |
| | 2018 | 16.3 | 0.005 | 0.009 | 0.010 | 1838 | 1899 |

4) Spatial autocorrelation of rainfall: using the mean of all hexagon gauges as the precipitation reference for the other type of gauges ignores spatial variance of precipitation within the hexagon area. There is no information about the size of the hexagon; maybe it is too small, and the variance is neglectable. Still, in principle, the difference between the lysimeter spatial average and the gauge measurement may be related to their spatial scale.

We added more information on the site experimental setup. Line 161 – 164: "The hexagon area covers approximately 49 m² in total. The precipitation gauges and weather stations are located directly adjacent (Fig. 2 B), except for one tipping bucket gauge in Rollesbroich, which is located around 30 meters from the respective lysimeter hexagon." Based on the short distances between the lysimeter themselves, the other measurement methods, and weather sensors, we do not expect a relevant influence of spatial autocorrelation on the rain data.

5) Uncertainty estimation: is computed based on the standard deviation between measurements for each hour. This computation is based on the assumptions: 1) the precipitation measurements (for a given hour) are normally distributed with the same mean and variance for all gauges (the text says the first part of this statement), and 2) the data from the different gauges are independent. How can these two assumptions be justified?

The uncertainty estimation based on the standard deviation was only done for the lysimeter data. Here, we calculated the mean between the aggregated hourly precipitation data for all available lysimeters during that hour.

We assumed that the deviations from this mean precipitation of the respective hour determined by each lysimeter are normally distributed.

Besides, we calculated the mean and variance for all lysimeter measurements at each site (Table 2; Appendix D1). We found that the lysimeters have a good agreement. This assumption is supported by good correlation of the lysimeter data shown in Appendix C.

**Table 2:** Mean and variance for all lysimeter measurements at each site for hours in which every lysimeter at a site has been available during the investigation period (2015 – 2018). Data filtered for "rain" and $P > 0.1$ mm h$^{-1}$.

| | Lysimeter | Rollesbroich | Selhausen | Dedelow |
|---|---|---|---|---|
| **Mean (mm h$^{-1}$)** | 1 | 0.887 | 0.824 | 0.815 |
| | 2 | 0.901 | 0.826 | 0.823 |
| | 3 | 0.907 | 0.824 | 0.829 |
| | 4 | 0.884 | 0.822 | |
| | 5 | 0.877 | 0.826 | |
| | 6 | 0.893 | 0.828 | |
| **Variance (mm h$^{-1}$)$^2$** | 1 | 1.67 | 1.27 | 1.93 |
| | 2 | 1.69 | 1.28 | 1.96 |
| | 3 | 1.69 | 1.28 | 1.97 |
| | 4 | 1.67 | 1.27 | |
| | 5 | 1.64 | 1.30 | |
| | 6 | 1.66 | 1.28 | |

6) The highest observed precipitation rate is 20 mm/h. Can you provide some info about this value so the reader would know what part of the precipitation rate distribution the analysis is covering?

We corrected the false statement about the highest observed precipitation rate at each site (line 277 - 281): "Each category comprises a span of 0.1 mm h$^{-1}$ for $P_{ref}$ between 0 and $P_{ref\_max}$, which was the maximum observed precipitation intensity during the observation period at each site ($P_{ref\_max}$: 23.21 mm h$^{-1}$ (RO); 12.91 mm h$^{-1}$ (SE); 32.24 mm h$^{-1}$ (DD))." The highest precipitation rate was observed in Dedelow with 32.24 mm h$^{-1}$. With this regard, we adapted Figure 3 accordingly, so that the uncertainty ranges are shown for each site individually. Furthermore, we added a table in Appendix B1-2 with information on the min, max, mean, median and quantiles (5, 25, 50, 75, 95 %) of the precipitation rate distribution at all sites for both, $P_{ref} > 0$ mm h$^{-1}$ and $P_{ref} \geq 0.1$ mm h$^{-1}$.

7) Section 2.8: Precipitation data corrections: This section proposes correction procedures that seem very empirical. How much can we trust these methods in the general case? How applicable are they for locations different than their development?

We agree that the correction methods are based on empirical data. The method after Richter has been developed especially for different locations with scarce data coverage. Therefore, the parametrisation is done loosely based on specific location characteristics. Here, the measurement method is important since the original method has been exclusively developed for Hellmann-type gauges. For the Dynamic Correction Method, factors for precipitation type, wind speed and gauge model are defined. The only site-specific factor appears to be the wind speed, which varies according to the site. Since the method has been developed covering a range of wind speeds including those measured at the sites, the factors should cover the issue.

We added the following sentence to the discussion (line 700 – 702): "Both methods were based on empirical data obtained at different locations than the test sites. However, the driving forces for data correction were wind speed and gauge design, which in this study were similar to those of the original data correction studies."

8) The following sentence appears in the conclusion section (L677): "The arithmetic mean of the lysimeter measurements has proven to be an almost unbiased reference for the precipitation measurement method". I don't think this was proved, but rather the assumption was the basis for the error analysis. Please clarify or correct.

We adapted the corresponding passage (line 736 – 737) in the revised manuscript: "The low bias in hourly lysimeter measurements indicated the suitability of their arithmetic mean as a reference for comparing precipitation methods."

9) L151 – what is the hexagon area?

We added an estimation of the hexagon area (line 161 – 164; "The hexagon area covers approximately 49 m² in total (Fig. 2A). The precipitation gauges and weather stations are located directly adjacent (Fig. 2 B), except for one tipping bucket gauge in Rollesbroich, which is located around 30 meters from the respective lysimeter hexagon."). We also changed the reference to Fig. 2A and added a scale and the approximated area to the figure.

10) L225: "iii) summing the minutely to hourly values" – do you mean the raw data or after the application of Eq 1,2, 3?

The procedure is done after the application of these equations. We revised the corresponding passage (line 240 – 243) as mentioned in response to comment 1).

11) Eq 4 + L247: "and $n_{ia}$ is the number of lysimeters with missing data during time interval i (-).". It is not clear to me what is the definition of nia here.

$n_{ia}$ are the lysimeters which have been inactive or did not provide reliable data during time interval I and thus have been set to "not available" during post-processing. We adjusted the terminology to $n_{na}$ and referred to non-available data (line 264-266).

12) L255: you should state that it is assumed the measurements are from a normal distribution with the same mean and standard deviation

The term "same mean and standard deviation" in in this context would be misleading since we calculate with one lysimeter data point per hour. Therefore, we assume that these values are normally distributed around the mean calculated by the three to six values provided by the lysimeter at each site. We changed the corresponding passage (line 274 – 275) to "It is assumed that the aggregated hourly precipitation amounts determined by all available lysimeters are normally distributed around $P_{ref}$."

13) It would be good to show the CDFs of hourly values of precipitation and ET.

We provided more information on the distribution of hourly precipitation values (Table 1; Appendix B), which can substitute for the CDFs for precipitation. Due to lack of data, we cannot determine the distribution of ET quantities.

14) Eq. 9: index i is missing

We added the missing index.

15) I think something is wrong with Eq. 10; please correct

We removed the excess superscript.

16) Eq. 8-10: Use either small or capital letters consistently.

We unified the respective equations and variables.

17) L350: Eq. 13 – should it be Eq. 14?

We adjusted the reference to the correct equation.

**Response to comments by Referee #2**

The manuscript of Schnepper et al. presents a comprehensive comparison of lysimeter precipitation measurements and four typical point measurements techniques, and suggests and evaluates two potentially promising correction procedures. The manuscript is well written and clear, and well suited for publication in HESS. Overall, this is an impressive dataset and an interesting study. However, the presentation of the extensive dataset could be improved to strengthen the overall output. Here are some suggestions that you might want to consider:

We thank anonymous Referee #2 for the constructive feedback and the suggestions made.

For the correlation analysis a major reduced major axis regression should be used, as you'd expect uncertainties on both the x and the y axis, while classic linear regression assumes that the x-axis has no error/uncertainty (see Harper 2016). To reduce the bias of the large number of small rainfall events at the sites you might want to use binning procedures instead of showing point clouds, this will help to better visualize if certain rainfall intensities are less / more affected by undercatch or the impact of wind speed on CR and how these change for different bins.

We have chosen to continue using the classical linear regression based on least squares (LS) that accounts only for uncertainties/errors on the y-axis. This decision is based on the assumption that the uncertainty of the reference precipitation is negligible compared to the uncertainties associated with the investigated precipitation measuring methods. Our analysis has revealed that for the weighing gauges, the regression line produced by reduced major axis (RMA) regression closely approximates the LS regression- and 1:1 reference line. However, for the acoustic sensors, the RMA regression line deviates considerably from the LS regression line towards a closer fit to the 1:1 reference line. This deviation attributes uncertainties/errors to the reference and is not easily explainable, as the reference data exhibits good correlations with the weighing gauges, laser disdrometers, and some tipping buckets. Therefore, the deviations from the 1:1 line observed with the data from the acoustic sensors are primarily attributed to uncertainties associated with the acoustic sensor measurements.

The use of binning to improve the visualisation of the correlation plots and reduce the bias of the large number of small rainfall events appears promising. However, we think that applying this method would also come with a loss of information, for example considering the scattering of the acoustic sensor data and other outliers. Additionally, binning would require selecting certain ranges for the respective bins. These ranges need to be large enough to provide sufficient details of the original data. Providing the statistics (bias, standard deviation) for the respective bins for each gauge would contribute to a lack of clarity. We decided to use binning for the wind speed analysis. More details are given below.

As pointed out by referee 1 the effect of ET during the 1-hour rainfall aggregates and the effect of NRE needs to be explained and potential uncertainties of ET need to be discussed in the revised version of the manuscript. Consider estimating NRE for your lysimeter sites and adding the total (annual) amount of NRE in table 5 or at a later point in the manuscript (I understand that there are uncertainties, however an estimate would be helpful).

We think that $ET_a$ does not affect the hourly atmospheric water input (AWI) data considerably, because the distinction between AWI and $ET_a$ was made on minutely basis. Therefore, only AWI or $ET_a$ are underestimated if AWI and $ET_a$ occurred during the same minute. Regarding non-rainfall water (NRW), we found that during sunset and sunrise, and with a threshold of 0.07 mm $h^{-1}$, around 17.3, 14.3 and 20.7 mm $a^{-1}$ have been deposited at Rollesbroich (Ro), Selhausen (Se) and Dedelow (Dd), respectively. This is equal to 2.3, 2.9 and 4.6 % of the average annual precipitation at the sites. These numbers must be interpreted carefully, since they heavily depend on data availability of the devices and assumptions on the formation of NRW. We added a table in Appendix E with data on NRW for each site. The table comprises total annual amounts of NRW, mean, median and standard deviation of NRW-events, number of events and number of potential NRW-events which exceed the assumed maximum water amount to deposit during an hour of NRW of 0.07 mm $h^{-1}$. The median for NRW at Ro, Se and Dd were 0.0085, 0.0068 and 0.0075 mm $h^{-1}$, which was equal to 2.05, 1.31 and 1.68 % of the median hourly precipitation at the respective sites. With the data obtained, we can assume that the reference precipitation of the lysimeter is slightly affected by NRW, but we do not think that this effect would distort the results of the gauge comparison.

The analysis of the effect of wind speed is not fully satisfying in the current version of the manuscript. Hourly wind speed averages might not be sufficient to thoroughly analyze the effect of wind on the CR. One (hopefully easy) sensitivity test could be replotting Figures 8 to 11 with e.g. 10min resolution data (the smaller the timestep

the more convincing) and discussing if and to which extent this changes the obtained results. One could show the 1h to 10min comparison for TB in the manuscript and leave the rest for the supplement. Please also consider the suggestions above to improve the presentation of results.

We tested plotting the CRs as functions of the wind speed with 10-min resolution. However, we did not find any improvements in extracting information of the effect of wind speed on the CRs. Applying binning, as suggested by Referee 2 in an earlier comment, helped to visualise wind speed effects more clearly. We used bins of 0.1 mm h$^{-1}$ and calculated the mean CR for each bin. We plotted the data similar to the figures used before but adjusted the y-axis to show a smaller range and the x-axis to be continuous instead of logarithmic (Fig. 8-11). The results were similar for 10-min and hourly data. For the sake of consistency, we sticked to the hourly data.

Here are some suggestions for minor changes in the abstract, I did not follow up with a detailed line to line edit throughout the rest of the manuscript. Overall, the manuscript is well-written and it was an interesting read. Sometimes sentences are a bit long and complicated, you might want to edit these when revising the manuscript (i.e., 183-185,…)

We shortened the mentioned (199-201) and further sentences in the manuscript (304-306, 353-355, 599).

L15: change "true" to actual

Adjusted the term in the manuscript.

L17: remove "under different climate conditions" -> your current data collection is limited in this regard (e.g., similar mean annual precipitation & temperatures, almost no snow events, heavy precipitation events)

Changed the passage to "[…] quantified with those obtained by lysimeters in different regions in Germany."

L23: acoustic sensors

Adjusted the term in the whole manuscript accordingly.

L24: rephrase: … 1-hour aggregated values above a threshold of 1mm h were compared.

We rephrased the sentence to "From 2015-2018, data were collected at three locations in Germany and 1-hour aggregated values for precipitation above a threshold of 0.1 mm h$^{-1}$ were compared."

L30: hourly measurement bias

Adjusted the term in the whole manuscript accordingly.

L34: rephrase: generally lead to recording lower precipitation amounts and L 35: rephrase: therefore might contain significant uncertainties.

We shortened and rephrased the sentences to: "The results indicate that considerable precipitation measurement errors can occur even at well-maintained and professionally operated stations equipped with standard precipitation gauges. This generally leads to an underestimation of the actual precipitation amounts."

L41 remove first sentence of introduction

We removed the first sentence of the introduction.

In Figure 1, can you highlight the three selected sites (only)?

We adjusted the figure accordingly to mainly highlight the three sites.

Consider adding a table with the main site characteristics (mean temperature, precipitation, coordinates, etc.)

We replaced the information on the main site characteristics provided in the text (line 136 – 150) with a table (Table 3).

**Table 3**: Information on site characteristics, annual mean precipitation (DWD, 2021a) and annual mean temperature (DWD, 2021b) based on multi-year averages (1961 – 1999).

|  | Mean annual precipitation (mm) | Mean annual temperature (°C) | Coordinates (N ° ', E ° ') | Dominant land use (-) |
|---|---|---|---|---|
| **Rollesbroich** | 1063 | 7.7 | 50°37', 06°18' | Grassland |
| **Selhausen** | 723 | 9.8 | 50°52', 06°27' | Arable land |
| **Dedelow** | 504 | 7.9 | 53°22', 13°48' | Arable land |

Add distances in Figure 2A.

We added a scale and marked the hexagon area in Figure 2A.

The hours of slight, moderate and heavy precipitation (Table 2) should add up to Rain + Mixed + Snow or Rain only (that would be my preferred option because you excluded the mixed and snow events right?) in Table 1. Currently the numbers do not add up. You could combine Table 1 & 2 in one table Rain (slight, moderate, heavy), Mixed, Snow.

We merged table 1 & 2 by showing the hours of slight, moderate, and heavy rain and all intensity ranges for mixed and snow (Table 4). Also, we ensured the consistency of the displayed values within the manuscript.

**Table 4:** Number of hours classified according to the reference rain intensity ($P_{ref}$) and number of hours with precipitation classified as "mixed" and "snow" for the years 2015–2018. Hours for which no reference could be calculated due to missing data and hours with $P_{ref} < 0.1$ are excluded.

| Site | Slight rain ($0.1 \leq P_{ref} < 2.5$ mm h$^{-1}$) (h) | Moderate rain ($2.5 \leq P_{ref} < 10$ mm h$^{-1}$) (h) | Heavy rain ($10 \leq P_{ref} < 50$ mm h$^{-1}$) (h) | Mixed ($0.1 \leq P_{ref} < 50$ mm h$^{-1}$) (h) | Snow (h) |
|---|---|---|---|---|---|
| **Rollesbroich** | 3338 | 257 | 1010 | 639 | 26 |
| **Selhausen** | 2675 | 160 | 4 | 146 | 9 |
| **Dedelow** | 2047 | 127 | 4 | 261 | 14 |